

**Economic impacts of drought risks for water utilities through**
**Severity-Duration-Frequency framework under climate change**
**scenarios**
Diego A. Guzmám[1-2], Guilherme S. Mohor[1], Denise Taffarello[1] and Eduardo M. Mendiondo[1]
[1] Department of Hydraulics and Sanitation - Sao Carlos School of Engineering, University of Sao Paulo - Sao
Carlos, SP, 13566-590, Brazil
[2] Department of Civil Engineering, Pontificia Bolivariana University, Bucaramanga, STD, 681007, Colombia
*Correspondence to*: Diego A. Guzman (daga2040@hotmail.com)
**Abstract**
Climate variability and increasing water demands prioritize the need to implement planning
strategies for urban water security in the long and medium term. However, actions to manage
the drought risk impacts entail great complexity, such as the calculation of economic losses
derived from the combination of severity, duration and frequency under uncertainties in the
climate projections. Thus, new approaches of risk aversion are needed, as an integrated
framework for resilience gap assessment, for water utilities to cope with droughts, thereby
linking drivers of climate, hydrology and human demands. This paper aims to present the
economic impacts of risk aversion for water utilities through a framework linking severity,
duration and frequency (SDF) of droughts under climate change scenarios. This new model
framework addresses the opportunity cost that represent the preparedness for risk aversion to
cope with potential future impacts of droughts, involving a set of options for planning of
water resources, under different demands and climate projections. The methodology
integrates the hydrological simulation procedures, under radiative climate forcing scenarios
RCP 4.5 and 8.5, from a regional climate model Eta-INPE, with time horizons of 2007-2040,
2041-2070, and 2071-2099, linked to Water Evaluation and Planning system (WEAP)
hydrologic model and under stationary and non-stationary water supply demand assumptions.
The model framework is applied to the Cantareira Water Supply System for Sao Paulo
Metropolitan Region, Brazil, with severe vulnerability to droughts. By using hydrological
simulations with WEAP, driven by Eta-INPE Regional Climatic Model base line scenarios
(1962-2005), were characterized the SDF curves. On the one hand, water tariff price
associated to calibrated and modelled scenarios constitute supply/demand proxies of the water
warranty time delimited by drought duration. Then, profit loss analysis scenarios are assessed





for the regional water utility. On the other hand, for drought resilience gap, results show water
utility profit losses per period between 1.3% and 10.3% of the regional GDP in 2016.
Although future economic impacts vary in a same order, non-stationary demand trends
impose larger differences in the drought resilience gap, when the future securitization are
linked to regional climate outputs.
**Key Words:** Climate change, Water Security, Severity-Duration-Frequency curves,
Economic profit losses
## 1.  Introduction
Climate change, population growth and uncontrolled urban/industrial development make
society more dependent on water (Montanari et al., 2013). The complex interaction between
meteorological, terrestrial and socio-economic water distribution schemes are the main factors
that define droughts (Lloyd-hughes, 2013; Van Loon et al., 2016a, 2016b; Wada et al., 2013).
Thus, to face a prospective drought scenario, with the demand as a determinant anthropogenic
factor requires society to rethink the way forward, mainly to reduce its vulnerability by
mobilizing more water for its use, by expanding and making use of alternative sources or by
regulating its demand (Falkenmark and Lannerstad, 2004; Kunreuther et al., 2013; Wanders
and Wada, 2015).
In terms of drought, a hydrological drought is defined as a negative anomaly in surface and
subsurface water levels (Van Loon, 2015; Wanders et al., 2017). These negative anomalies on
the surface, related to a level of water demand can cause water systems to collapse and trigger
strong socioeconomic impacts or the so-called socioeconomic drought (Mehran et al., 2015).
Droughts may not be as apparent as floods, but have proven to be one of the most complex
risks due to their slow development, strong and long lasting impacts such as broad geographic
coverage (Bressers and Bressers, 2016; Frick et al., 1990; Smakhtin and Schipper, 2008; Van
Lanen et al., 2013). Furthermore, various studies have shown that more severe and prolonged
droughts are expected for the future, leading to greater economic consequences,
environmental degradation and loss of human lives (Asadieh and Krakauer, 2017; Balbus,
2017; Berman et al., 2013; Freire-González et al., 2017; Prudhomme et al., 2014; Shi et al.,
2015; Stahl et al., 2016; Touma et al., 2015).  Therefore, it is essential to create appropriate
expectations about their potential impacts, aiming to mitigate catastrophes, reduce the risks of
damage and build a more resilient community (Bachmair et al., 2016; Mishra and Singh,
2010; Nam et al., 2015).





The need for a broader perspective in terms of comprehending and managing the impacts of drought requires actions to integrate their states or categories (Hao and Singh, 2015; Van Loon, 2015). This implies in studying droughts, understanding their propagation from meteorological phenomena, underground-surface dynamics and alterations of anthropogenic origin such as storage (Huang et al., 2017; Van Loon et al., 2016b; Wong et al., 2013). However, the most visible impacts on the water supply, energy generation, transport, recreation and water quality are strongly related to hydrological drought and not directly to meteorological drought (Van Lanen et al., 2016). Thus, we in this work addresses hydrological droughts as the main driver of direct economic impacts when water demand exceeds supply (Bressers and Bressers, 2016).

The availability of water supply new sources every day are more scarce, so the demand regulation is a strategy that is being considered by the supply companies to guarantee reliability during the drought (Zeff and Characklis, 2013). Among the demand control strategies are price-based policies ones, these seek to change the user's consumption pattern based on economic penalties or incentives (Millerd, 1984). However, the implementation of these strategies entails a great complexity in their planning and a high risk of utility losses for the water company.

The São Paulo Metropolitan Region (SPMR) located in the south east of Brazil, which has approximately 20 million inhabitants, is an important economic center in Latin America that influences approximately 12% of the Gross Domestic Product (GDP) in Brazil (Haddad and Teixeira, 2015). During the (2013-2015) period, the population of the SPMR experienced a significant reduction in water resources availability and decrease in the water supply (Coutinho et al., 2015; Nobre and Marengo, 2016; Taffarello et al., 2016b). Consequently, the 2013-2015 water deficit had socioeconomic impacts such as widespread social protests, increases in food prices and energy tariffs in homes, industries and commerce (Hanbury, 2015). The Federation of Industries of the State of Sao Paulo (FIESP) estimated that 60,000 establishments, which represent almost 60% of the state's industrial GDP, are affected by a lack of water (Marengo et al., 2015). In addition, from 2014 to 2015, the Sao Paulo State Water Utility Company (SABESP) recorded an average annual liquid net income reduction of approximately 75% compared to 2016, leading to a major financial crisis in the company (GESP, 2016). Thus, as long as there are no systematic and detailed studies on the impact of drought on the regional economy, shaping financial planning policies is a complex and uncertain task that must be reinforced. Based on the severity and duration of the water deficit,



this article aims to assess the economic impacts of drought risks for water utilities through
integrating a severity-duration-frequency framework under climate change scenarios. Also,
this paper describes an academic exercise to manage drought financial planning for the
SPMR, considering the perspective of the economic impact on the Sao Paulo Water Utility
company.
The sections of this article outline interconnected methods and criteria, explained as follows.
In Section 2, the text describes the study area (see Figure 1) and water crisis
contextualization. Section 3 outlines the methodological approach starting with the
hydrological modeling, characterization of the droughts using the threshold level method, the
formulation of the SDF curves of the system and subsequently the links climatic, hydrological
and economic aspects of the methodology (Figure 2). In Section 4, the results obtained are
shown as financial drought planning scenarios. Finally, in Section 5, the discussion and
conclusions are presented regarding the proposed approach.
**2.   Study area and water crisis contextualization.**
The Cantareira Water Supply System, hereafter referred to as the Cantareira System, is
located in the South-East of Brazil between the states of Sao Paulo and Minas Gerais (-46.9 to
-45.7 longitude and -22.5 -23.5 latitude). The regional climate is classified as subtropical –
sub-humid, with a maximum annual average temperature of 25 °C and a minimum annual
average of 15 °C (Blain, 2010; Rodríguez-Lado et al., 2007). On the other hand, the rainfall in
the Southeast of Brazil presents an annual cycle, with maximum rainfall from December to
February (summer) and minimum rainfall from June to August (winter). The rainy season in
the Cantareira System generally begins at the end of September and ends in March. In this
period, on average 72% of the rainfall in the region is accumulated (Marengo et al., 2015). In
hydrological terms, 2265 km$^2$ of drainage area into the system historically generates an annual
mean tributary discharge of 38.74 m$^3$/s, where the Jaguarí tributary contributes approximately
46%. Structurally, the system consists of the damming and interconnection of five basins with
a useful total storage volume of 988.8 hm$^3$, arranged to transfer water from the Piracicaba
River Basin to the Upper Tietê Basin (Fig. 1). Thus, the system had been configured to supply
water to about 8.8 million people in the SPMR  before the last acute crisis in 2013-2015 (De
Andrade, 2016; Marengo et al., 2015; Nobre et al., 2016; Nobre and Marengo, 2016;
PCJ/Comitês, 2016, 2006).
Previously in the SPMR, some water shortages were recorded. The first one was during 1953-
1954, then from 1962 to 1963 (Nobre et al., 2016), which apparently motivated the





construction of the Cantareira system and the latest one was from 2000 to 2001 (Cavalcanti
and Kousky, 2001). Thus, the system, designed to supply the increasing demand for water in
the SPMR, began its partial operation in 1974 and its construction was completed in 1981
with a 30-year permit to transfer up to 35 m$^3$/s according to a periodic technical report (Mohor
and Mendiondo, 2017; Taffarello et al., 2016a). Cantareira System is currently administered
by SABESP, which mainly operates the water network in the SPRM, and with Government of
the State of Sao Paulo as its main shareholder.
However, various studies have identified changes in trends in rainfall and temperature
extremes, showing an increase in the intensity and frequency of days with heavy rainfall and
longer duration of hot dry periods between rainfall events in South America and southeastern
Brazil (Chou et al., 2014b; Dufek and Ambrizzi, 2008; Haylock et al., 2006; J. A. Marengo et
al., 2009; Jose A. Marengo et al., 2009a, 2009b; Nobre et al., 2011; Zuffo, 2015). Although
historically, the SPRM study area is not affected by droughts of the same order of Northeast
Brazil, the SPRM is progressively becoming vulnerable to water shortages. Therefore, during
the recent period of the acute crisis 2013/2015, SABESP were taken reactive measures, to
control the consumption in the SPMR, such as (Marengo et al., 2015): Programmed water cut-
offs; Bonuses and penalties to reduce and increase consumption, respectively; Extraordinary
increases of water tariff cost; Network pressure reduction; Water use from the reservoirs´
dead volume; Social awareness campaigns to inform people about shortages; Water
distributed by tankers in the most critical areas of the city to provide the Basic Water
Requirement (BWR) for human needs. Nevertheless, according to SABESP, there is currently
a gradual system recovery, which enables the reestablishment of pre-crisis supply levels
(SABESP, 2016a).

**3.  Methodology**
The methodology was followed in three modules that are summarized in Figure 2. In the first
module, the hydrological simulation was approached by the Water Evaluation and Planning
tool (WEAP) (Yates et al., 2005a). The model was calibrated and validated, based on the
available historical hydrometeorological information (2004-2015) for the study area. Then,
from the calibrated hydrological model and the RCM Eta-INPE historical periods datasets, the
base discharge scenarios were estimated. In the second module, in the TLM approach, the
"threshold" had to be defined according to stationary and non-stationary assumptions of water
demand in the SPMR. Afterwards by analyzing the duration series and extreme deficits
through GEV (Generalized Extreme Value) distribution, the Severity-Duration-Frequency





curves (SDF) were developed (J. H. Sung and Chung, 2014). To complete the second module,
the average water price is defined per each cubic meter of deficit, as a function of the supply
warranty time during the hydrological drought events, to configure the baseline analysis
scenarios. The final module evaluates through the baselines scenarios the Water Utility
Company economic profit losses, under the hydrological model WEAP output datasets,
driven by the Eta-INPE, RCPs and (2007-2040, 2041-2070, 2071-2099) scenarios, previously
processed by the TLM approach. It should be clarified that, for the analysis under the non-
stationary assumption, the growth of water consumption is represented in each projection time
step, that is, to 2005-2040 correspond 31 m$^3$/s, to 2041-2070 correspond 38 m$^3$/s and to the
period 2071-2099 correspond 43m$^3$/s.
The results of the methodology of Figure 2 can be seen as the opportunity cost, which would
represent appropriate financial planning, considering the anticipation of drought events by
implementing adaptation measures, supported economically by the forecast of the potential
impacts. These impacts are shown as a set of potential scenarios involving climate
uncertainty, human triggering factors and the prediction of extreme theory (Baumgärtner and
Strunz, 2014; Wanders and Wada, 2015). Thus, the approach seeks to provide a planning
water-security support analysis in areas highly dependent on surface water resources.
As a complement to Figure 2, the main variables that induce the change scenarios for this
study are shown in Table 1.
**3.1. Climate and hydrological modeling**
Currently the RCM Eta-INPE (Brazilian National Institute for Space Research) plays an
important role in providing information for local impact studies in Brazil and other areas in
South America  (Chou et al., 2014b). In order to assess the uncertainties of climate change
impacts, the simulation results of the Eta-INPE model were used in this paper. The model is
nested within the GCMs MIROC5 and HADGEM2-ES, forcing by two greenhouse gas
concentration scenarios (RCPs) 8.5 and 4.5 [W/m$^2$] used in AR5 (IPCC 5th Assessment
Report); with a horizontal grid size resolution of 20 km x 20 km and up to 38 vertical levels
through 30 years of time slices (periods) distributed as follows: 1961-2005 (as the baseline
period), 2007-2040, 2041-2070 and 2071-2099 (Chou et al., 2014a, 2014b; Prudhomme et al.,
2014). The climate projections of the Eta-INPE model was used to drive the WEAP Rainfall
Runoff Model-soil moisture method (World Bank, 2017; Yates et al., 2005a). The WEAP,
developed by the Stockholm Environment Institute US Center, is an integrated water resource



planning tool used to develop and assess scenarios that explore physical changes (natural or
anthropogenic) and has been widely used in various basins throughout the world (Bhave et al.,
2014; Esteve et al., 2015; Groves et al., 2008; Howells et al., 2013; Mousavi and Anzab,
2017; Psomas et al., 2016; Purkey et al., 2008; Vicuna and Dracup, 2007; Vicuña et al., 2011;
Yates et al., 2005b). Climate-driven models, such as WEAP provide dynamic tools by
incorporating hydroclimatological variables to analyze, in this case, a one-dimensional, quasi
physical water balance model, which depicts the hydrologic response through the surface
runoff, infiltration, evapotranspiration (Penman-Monteith equation), interflow, percolation
and base flow processes (Forni et al., 2016).
The hydrological model comprises 16 sub-basins with a spatial resolution ranging from 67 to
272 km$^2$ (see Table A-1 in supplementary material - section A), which defines the natural
discharge produced by the Cantareira System. The observed hydrologic data (discharge and
rainfall) were taken from HIDROWEB (the National Water Agency database [ANA]),
SABESP and the São Paulo state Water and Electricity Department [DAEE]. A network of 52
rain gauge stations and 11 discharge gauge stations were configured, with inputs and outputs
by a monthly time-step. On the other hand, the meteorological data from 14 gauge stations
(temperature, relative humidity, wind speed and cloudiness fraction) were taken from the
National Institute of Meteorology and Center for Weather Forecasting and Climate Research
(CPTEC) databases. For the basin characterization, we adopted the soil map from (De
Oliveira et al., 1999) (1:500,000) and the land use map of 2010 from (Molin et al., 2015)

217  (1:60,000).

The WEAP model was calibrated using an automatic PEST tool module (Doherty and Skahill,
2006; Seong et al., 2015; Skahill et al., 2009; Stockholm Environment Institute (SEI), 2016)
and manual techniques on a monthly basis. In the modeling process, a two-year warm-up
period from 2004 to 2005 was established, for the calibration period from January 2006 to
December 2010 and from January 2011 to August 2015 as the validation period.  During this
process, the following variables were calibrated: Kc (Crop Coefficient), SWC (Soil Water
Capacity), DWC (Deep Water Capacity), RZC (Root Zone Conductivity) and PFD
(Preferential Flow Direction). The chosen performance criteria, widely used in hydrologic
applications, were the Volumetric Error Percent Bias (PBIAS), Standard Deviation Ratio
(SDR), Nash-Sutcliffe Efficiency (NSE), NSE of the logarithmic of discharges (NSELog)
which is more sensitive to low-flows, Coefficient of determination (R$^2$) and Volumetric
Efficiency (VE) criterion (Muleta, 2012).





The calibration and validation procedure of the hydrological model was carried out from
upstream to downstream streams with historical discharge information (refers to the reservoirs
inflows) from collected from ANA-HYDROWEB (www.ana.gov.br). Cantareira's reservoirs
were set up as a single Equivalent System (ES), where the specific water demands are adapted
(ANA and DAEE, 2004; PCJ/Comitês, 2006). This ES can be expressed as follows:
$ES_{Cantareira} = \sum_i^n QN_i - \sum_i^n WD_i$  Equation 1.
where $ES_{Cantareira}$ is the available water for withdrawal from the system, $QN$ is the natural
discharge from the reservoir $i$ and $WD$ is the specific water demand in each reservoir (such as
the Piracicaba River demand).
It is worth noting the sub-basins areas are smaller than each cell of the adopted climate model
(400 km²). Therefore, in order to adjust the dataset, the projections of the Eta-INPE scenarios
had to be adapted from/to the original location of the gauge station, and corrected according
to the observed historical climate conditions. The climate projections from Eta-HadGEM2-ES
and Eta-MIROC5 under RCP 4.5 and 8.5 scenarios were used in the hydrologic model to
evaluate the impacts and climate uncertainty in the discharge regime. The results can be seen
in supplementary material – section B (Fig. B-1) and are represented as future time slices of
30 years approximately: 2007-2040, 2041-2070 and 2071-2099, under the intermediary
(pessimistic in this study) and optimistic RCP scenarios (IPCC, 2014).
**3.2. SDF curve development**
Following the flowchart of Figure 2, the Threshold Level Method (TLM) is traditionally used
to estimate hydrological drought events from continuous discharge time series. TLM was
originally called 'Crossing Theory Techniques" and it is also referred to as run-sum analysis
(Hisdal et al., 2004; Nordin and Rosbjerg, 1970; Şen, 2015). Usually different values are used
to define the threshold in hydrological drought analysis by the TLM approach (Tosunoglu and
Kisi, 2016). In this study, two demand scenarios, approached as "threshold levels", were used
in the mean daily-monthly discharge data. Initially, a stationary demand of 31 m³/s was
defined as the historical average demand and another non-stationary demand of 31 to 42 m³/s
over time was defined as a hypothesis representative of the population growth in the SPRM
(see Figure 3). These water demand values are consistent with the ANA/DAEE, 2004 study,
according to the record and projection scenarios of the population growth of the IBGE[1].

---

[1] Brazilian Institute of Geography and Statistics: http://www.ibge.gov.br/home/

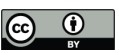



Based on the time series of "severity" (or deficit, in m$^3$) and duration (days) in the Cantareira
System, obtained from the hydrological modeling of the historical scenarios from the Eta-
INPE model (1962-2005), the SDF curves were constructed. To estimate the return periods of
drought events of a particular severity and duration, the block maxima GEV frequency
analysis distribution was used. In this case, the GEV distribution is useful because it provides
an expression that includes all three types of extreme value distributions (Tung et al., 2006).

In various studies addressing SDF curve development, the GEV distribution was consistent
with the data sets of extremes, where distributions that use three parameters were required to
express the upper tail data (J H Sung and Chung, 2014; Svensson et al., 2016; Todisco et al.,
2013; Zaidman et al., 2003). On the other hand, it is suggested that for other durations of
drought, other probability distribution functions can be explored (Dalezios et al., 2000;
Razmkhah, 2016). However, in this study we took advantage of the versatility of the GEV
distribution, considering its flexibility to fit a set of data through the expressions:
$$F(x) = exp\left[-\left\{1 + \xi\left(\frac{x-\mu}{\sigma}\right)\right\}^{1/\xi}\right] \quad \xi \neq 0 \qquad \text{Equation 2.}$$
$$F(x) = exp\left[-exp\left(-\frac{x-\mu}{\alpha}\right)\right] \qquad \xi = 0 \qquad \text{Equation 3.}$$
where the cumulative distribution function $F(x)$ depends on $\mu$ as a location parameter, $\alpha$ as a
scale parameter and $\xi$ as a shape parameter. Therefore, if, $\mu+\alpha/\xi \leq x \leq \infty$ for $\xi < 0$ is a Type
III (Weibull), $-\infty \leq x \leq \infty$ for $\xi = 0$ is a Type I (Gumbel), and $-\infty \leq x \leq \mu +\alpha/\xi$ for $\xi > 0$ is a
Type II (Frechét) distribution (Stedinger et al., 1993).
In order to fill a considerable number of events per interval, droughts were classified into four
time intervals 31, 90, 180 and up to 365 days. Thus, considering the adoption of the GEV
distribution, the model parameters $\xi$, $\alpha$ and $\mu$ for cumulative durations defined and return
periods of 2, 10 and 100 years were estimated using the Method of Maximum Likelihood
Estimator (MLE).The SDF curves of the hydrological drought in the Cantareira System are
shown in Figure 4. In addition, the adjusted parameters table and the diagnostic plots QQ-plot
and Return Level vs. Return Period for the GEV distribution can be seen in the supplementary
material as sections C and D.





### 3.3. Water price and Hydrological drought relationship

According to the flowchart of Figure 2, drought can be addressed as a somewhat unusual economic phenomenon in that it affects both supply (the source) and demand (users), especially in systems dependent on water from a single source (Moncur, 1987). As expected, episodes of water scarcity pose technical, legal, social and economic problems for managers of urban water systems. Traditionally to overcome these episodes, reservoirs play a key role in water supply and demand management, providing security against hydrological extremes (Mehran et al., 2015). However, when the water deficit intensifies, the structural measures are not enough and they must be accompanied by contingency measures.

In recent years, the Cantareira System played an important role to guarantee the water supply in the SPMR. Figure 5 shows the TLM analysis with a constant threshold under two discharge scenarios, a) monthly natural discharge and b) regulated discharge, where the regulated discharge is represented by the annual average aggregation of monthly natural discharges. Thus, without the reservoirs, i.e. withdrawals dependent on the instantaneous inflow, the average accumulated deficit over these 17 years would be 225% greater. Considering this assumption, the analysis showed two hydrological drought periods in 2000-2003 and 2010-2015 (Figure 5b); one with a lower and another with a higher deficit, respectively. While for the period from 2004 to 2009, a series of smaller droughts in both magnitude and frequency could be overcome by the reservoir system. On the other hand, in 2010-2015, the accumulated deficit, under the regulated scenario, would exceed the useful storage in 70%; while for the period 2000-2003, the accumulated deficit only reached 43% of the system's useful storage capacity. Therefore, it is clear that over a long period of deficit or strong multi-year droughts, the system of storage could be accompanied by contingency complementary measures.

Urban drought management programs incur costs that must be assumed to overcome the water crisis with equity (Molinos-Senante and Donoso, 2016). SABESP in the SPMR, for example, through price-based policies controlled the consumption rates of water users when the hydrological deficit scenarios were presented in the Cantareira System (Millerd, 1984, SABESP, 1996, Iglesias and Blanco, 2008), see Figure 6. Thus, during the 2014/2015 drought in SPRM, reactive economic contingencies were implemented, such as increased water tariff costs, extra fees and price incentives, which had a detrimental effect on the company's profit margin, which provides the water resource. (GESP, 2016). Although the relationship between the Water Deficit and the tariff Adjustment Rate show a relatively low Pearson correlation



coefficient "$r_{xy}$" of 0.398, this may be useful given the lack of information regarding drought
and its economic impacts on the study area.
In Brazil, each state-owned sanitation company has its own water charging policy, where the
vast majority use block tariffs as a pricing policy, including SABESP (De Andrade Filho et
al., 2015; Mesquita and Ruiz, 2013; Ruijs et al., 2008). In Sao Paulo State, the tariff policy
system is regulated by Decree 41.446/96, also for services provided by SABESP. For the
water tariff setting, several factors are taken into account, such as service costs, debtors
forecast, expenses amortization, environmental and climatic conditions, quantity consumed,
sectors and economic condition of the user (SABESP, 1996). These sectors are divided into
residential, industrial, commercial or public, and the value that is charged for the service is
always progressive. In other words, there is a standard minimum consumption with a fixed
value and, based on that, such factors vary the consumption ranges (SABESP, 2016b). From
the total water withdrawn from the Cantareira System, urban use is predominant in SPRM,
where approximately 49% of the total is for household needs, 31% for industrial needs and
20% for irrigation (Consórcio/PCJ, 2013). In this study, we consider the water-withdrawal for
domestic and industrial use in the SPMR, because the direct dependence of these sectors on
the SABESP water supply network, as well as the supply priority that these sectors have
according to Brazilian law.
The water price formation study is not part of this work as it entails a complex
microeconomic analysis, due to the diversity of variables in the process (Garrido, 2005).
Additionally, the financial exposure does not always exhibit a strong correlation with weather
indices (Zeff and Characklis, 2013). Therefore, in order to establish a water appraisal for the
economic analysis, an empirical relationship between the water tariff and its availability
according to the drought duration was developed. For this, the TLM analysis here presented
was performed from the monthly discharge series during from 2000 to 2016 (Figure 6a),
aiming to associate the resulting information with the previously obtained SDF curves. Thus,
the top part of Figure 7 shows the drought duration and the annual tariff adjustment with a
Pearson correlation coefficient "$r_{xy}$" of 0.402 between them, while the lower part represents
the volume deficit for each drought duration. Based on Figure 7, it can be observed that from
greater drought durations and deficits, there is expected an increase in the water tariff for the
following period. On the contrary, the smaller deficits are overcome with the water stored in
the system and the increase in tariffs is a consequence of the annual Consumer Price Index
(CPI) and other tariff updates according to the law.





According to the relationship established between the drought duration and the tariff
adjustments, assigning the average water price for this study requires some additional
assumptions explained as follows: (i) based on the current average rates for the domestic and
industrial sectors that range from US\$ 2.27 to US\$ 4.48 per m$^3$, respectively (SABESP,
2016c), an average price was established for the analysis of US\$ 3.38 per m$^3$, assuming that
this value is given considering normal supply conditions, (ii) from the four intervals of
drought duration considered for the SDF curve construction and the water tariff adjustments
of the analyzed period (min. 3.15% to max. 18.9%, see Figure E-1 in supplementary material
- section E), the water prices were established as a function of the drought duration by the
"supply warranty time percentage" as shown in Table 2.
Based on the assumptions shown in Table 2, the demand curve for the Cantareira System was
constructed as a function of the supply warranty time percentage (Figure 8). In this demand
curve, the reservoir network is considered to ensure water supply and provides resilience
during droughts of smaller magnitudes and duration. Overall, the curve represents the
inelastic behavior of the Price Elasticity of Demand (PED); showing closer intervals as water
supplies are reduced due to drought and higher prices imposed to try to reduce demands.
Hence a successful price-based rationing policy, requires a progressive increase if the demand
becomes predominantly inelastic (Mays and Tung, 2002), as the proposed hypothesis
establishes in this case. More studies of price elasticity and water scarcity can be found in
(Freire-González et al., 2017; Mansur and Olmstead, 2012; Ruijs et al., 2008).
From the drought events studied, i.e in 2000/2001(Cavalcanti and Kousky, 2001), in
2014/2015 (Nobre et al., 2016), which significantly affected the water supply, the TLM
analysis showed the interdependence between annual events (Figure 6b). Consequently, the
main impacts derived from water supply problems in the SPRM appear to be related to multi-
year drought events and medium to high severity such as the recent event. Therefore, based
on the 2000/2016 drought severity-duration-rate adjustment scenarios, three water supply
warranty scenarios were established (see Figure 8): 100% water availability, water availability
with storage dependency and water deficit with extra fees and other savings measures as a
good management practice to prevent strong impacts.
Thus, the baseline scenarios were configured to estimate the projections of the loss of
economic profits in the water utility company, due to the financial cost of the drought periods.
These scenarios are represented by the Severity-Duration-Impact curves, which are shown in





Figure 9, under different recurrence events, climate projections and demand variability
scenarios. Each pair of lines in Figures 9 a. b. (continuous and dashed) show the range of
uncertainty associated with the considered change variables.
The final step of the methodology (see Figure 2) calculated the impacts in terms of the
drought financial planning through the management horizons (2007-2040, 2041-2070 and
2071-2099). This calculation was carried out for the cumulative drought duration periods
greater than 180 days, considering that from this duration, the supply begins to show
important dependence of the Cantareira reservoir System.

## 4.   Results and discussions

The results section will be divided into: (i) hydrological modeling, (ii)   SDF curves and (iii)
economic results under climate changes.

### 4.1. Hydrological modeling

The hydrological model structure performed in monthly time steps, calibrated and validated
following a manually and automatic procedure. To improve the calibration procedure,
multiple statistical evaluation criteria were used, aiming to reduce the specific bias of any of
these, given the characteristics of the modeled series (Kumarasamy and Belmont, 2017). The
performance criteria of calibration and validation periods are shown in Table 3. The colors in
the Table 3 represent the classifications suggested by (Moriasi et al., 2007) and are as follows:
green for "very good" (NSE > 0.75; PBIAS < ±10%; RSR < 0.50), yellow for "good or
satisfactory" (0.75 > NSE > 0.5; ±10% < PBIAS < ±25%; 0.50 < RSR < 0.60), red for
"unsatisfactory" (NSE < 0.5; PBIAS > ±25%; RSR > 0.70).   Moreover, the correlation
coefficient ($R^2$) and the VE criterion values close to 1.0 mean that the prediction dispersion is
equal to that of the observation (Krause and Boyle, 2005; Muleta, 2012). Additionally, the
hydrographs for calibration and validation periods are shown in Figure 10. It is important to
note that in the validation period (2011-2015), part of the recent drought event was simulated.
Individual watershed hydrological modelling performance ratings are presented in
supplementary material - section A, Table A-1; also several statistical criteria were considered
to evaluate the calibration process, where each criterion covers a different aspect of the
resulting hydrograph. This is important because analyzing multiple statistics can provide an
overall view of the model based on a comprehensive set of indexes on the parameters
representing the statistics of the mean and extreme values of the hydrograph (Moriasi et al.,
2007). Five basins were modeled within the Jaguarí-Jacareí sub-system (Sub B-F28, B-F23,





B-F25, Jaguarí and Jacareí). This sub-system represents approximately 46% of the total
available water and showed the best modelling performance statistics, compared to the other
subsystems.

**4.2. SDF curves**

Using the traditional frequency analysis, the severity-duration-frequency curves for two
threshold levels and two RCMs discharge outputs were developed as shown in Fig. 4. For the
SDF curves configuration, the Generalized Extreme Values (GEV) function was used. Thus,
from the SDF results it can be observed that:
According to the fit data set (supplementary material - section C), the shape parameter ($\xi$)
varies with the drought duration, therefore for a drought interval of more than 180 days, the
Probability Distribution Function (PDF) Type I presents a better fit, even for the two
proposed demand scenarios. On the other hand, droughts with duration intervals of less than
90 days, under stationary and non-stationary demand scenarios, had a better fit to FDP Type
III (see Tables D-1 to D-4 in supplementary material - section D). Moreover, the fit diagnostic
plots "Empirical quantile vs Model quantile" (QQ-plot) and "Return level vs Return period"
(RR-plot) show the relationship between the model, the data fit and prediction capacity
(supplementary material - section C). Thus, in terms of the quantiles, the QQ-plot shows the
data trend to follow the model line in most cases. While the predictive capacity of the model,
represented by the RR-plot, shows a decrease as the return period increases.

**4.3. Economic impacts under climate change**

Based on the methodological approach (see Figure 2), the potential economic impacts were
calculated, produced by hydrological droughts greater than 180 days. These impacts are
presented considering the climate, demand, time and recurrence scenarios. Thus, the net
present value (NPV) of the economic detriment to the water utility company and the
percentage difference (Dif. %) between the demand scenarios are shown in Tables 4, 5 and 6
for each period.

From the results in Tables 4, 5 and 6, it can be observed that the economic impact is higher
for higher return periods as well as the step of stationary demand to non-stationary demand, as
expected. In addition, it is not possible to observe a differentiated trend in the results, when
they are forced by two different radiative scenarios over time. However, the scenarios nested
within HadGEM-ES, on average, presented lower values or with less economic impact, when
compared to the nested scenarios within MIROC5. Overall, the loss of economic profit from



2041 to 2070 showed lower values compared to the other two periods analyzed, probably due
to a more optimistic climate scenario in terms of surface water availability.
In Figure 11a, the box plot shows the dispersion of the economic impacts grouped under each
climate model by time periods. Results related to the MIROC5 model present a greater
dispersion than those related to the HadGEM-ES model. In this case, the upper extreme
values are related to the MIROC5 model, while the lower extreme values are similarly
distributed between the models. On the other hand (Figure 11b), the difference, in percentage,
related to the MIROC5 model show higher magnitudes and more stable differences over time
than those related to the HadGEM-ES model, denoting an impact-driven differentiation
between climatic models. Moreover, it can be observed in Figure 11 that, in response to the
growing projected demand, it will be expected an increase, in terms of the average percentage
of differences, for different time periods and for both climatic models.
In general, these results show the high complexity of the SPRM's drought risk and the
fragility of local GDP heavily dependent on water for their development. In the specific
impacts on the company's economy, the results showed losses per period between US$ 7929
and US$ 64582 million; these values, compared to the Gross Domestic Product (GPD),
represent an amount of between 1.3% and 10.3% of the last GDP in the state of São Paulo in
2016. As a consequence, the direct economic impacts on the water utility company, added to
other inherent problems to water shortage, can lead to a financial crisis with serious
repercussions in local economies.

## 5.  Conclusions and recommendations

This paper developed a methodology with application to assess economic impacts of drought
risks for water utilities through a framework under climate change scenarios. The SDF
framework has linked climate, hydrology and economy factors, using Sao Paulo Metropolitan
Region dependence on the Cantareira Water Supply System, Brazil. In this paper, we consider
these results preliminary, but with valuable information for a water utility interested in the
drought risk losses. Thus, the expected profit loss over the long-term would serve as the initial
estimate for financial contingency arrangements as insurance schemes, or community
contingency funds. In general, the SDF framework here developed can be proposed as a
planning tool to mitigating drought-related revenue losses as well as being useful for
development of water resource securitization strategy in sectors that depend on water to
sustain their economies.



Methodologically, first we characterized the hydrological droughts through the SDF curves, from the hydrological modeling by the baseline period of the RCM. Second, the SDF was coupled with a local water demand development based on the supply warranty time percentage during the drought events. Under these assumptions, an empirical drought economic impact curve was setup, representing the Water Utility Company profit losses due to the impossibility of supplying demand during hydrological drought periods. Additionally, our results could elicit further implications for drought risk reduction and management.

On the one hand, this SDF framework could help analyzing the impacts from key drivers, like climate, land use and water withdrawal rates in complex or recurrent drought patterns. Also, this SDF framework could couple interdisciplinary studies, with better relationships towards the nexus of water security, energy security and food security. Thus, we recommend future research of SDF framework linked to: Palmer's drought indices (Rossato et al., 2017), model-based framework to disaster management (Horita et al., 2017), ecosystem-based assessment for water security modeling (Taffarello et al., 2017), effectiveness of drought securitization under climate change scenarios (Mohor and Mendiondo, 2017). Moreover, SDF framework is capable of integrating actions towards: dynamic price incentive programs related to wise human-water co-evolution patterns, water-sensitive programs under deep cultural features, socio-hydrological observatories for water security, feasibility analysis of the economic impacts of implementing new technologies for water economy and flow measurement, leakage control, detecting and legalizing illegal connections and water reuse, among others. Furthermore, dissimilarities pointed from climate scenarios (see i.e. Figure 11) would suggest a set of possibilities to face the uncertainty. For instance, that SDF framework would guide the decision-making of water utility profits to cope with economic impacts of drought risks in long and medium term.

For further studies, it should be considered: that despite having achieved an acceptable performance, the inclusion of more gauge stations could not only improve calibration performance but also cover a larger sample space of events, increasing the confidence of projections. On the other hand, in order to have a methodological comparative standard, more regional studies of SDF curves need to be implemented, considering the spatialized analysis and broader statistics methods. Finally, it is a fact that the reliability of SDF curve estimates depends on the quality and extent of the records used, or in this case, the capacity of regional climate models to reproduce the observed distribution of extreme events.





**Acknowledgments**
The authors thank the support of several agencies of Brazil and Colombia: the Administrative
Department of Science, Technology and Innovation (COLCIENCIAS) Doctoral Program
Abroad, CAPES-PROEX-1650/2017/23038.013525/2017-30, CAPES Pró-Alertas
#88887.091743/2014-01, CNPq #307637/2012-3, CNPq #312056/2016-8 PQ and CNPq
#465501/2014-1 and FAPESP 2014/50848-9 Water Security of the INCT-Climate Change II.
The Sao Paulo State Water Utility Company, SABESP, kindly provided relevant information
for this study. All co-authors declare no conflict of interest.

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



**Tables.**
**Table 1.** Description of variables

| Scenarios' variables | Description |
|---|---|
| RCM Scenarios | Eta-Model nested in the GCMs:<br>· MIROC5<br>· HADGEM2-ES |
| RCP Scenarios [$W/m^2$] | Forcing by two greenhouse gas concentration scenarios:<br>· 4.5 as optimistic scenario<br>· 8.5 as pessimistic scenario |
| RMSP Water Demand Scenarios [$m^3/s$] | · Stationary Demand (SD) 31 $m^3/s$<br>· Non-Stationary Demand (NSD) 31 to 42 $m^3/s$ |
| Return period analysis Scenarios [Rp] | Rp = {2, 10, 100} years; drought severity (deficit $m^3$) and duration (days) scenarios. |


**Table 2.** Main assumptions for establishing the tariff water price according to the drought duration.

| Drought Duration Interval (days) | Water Tariff Adjustment adopted (%) | Average price (US$/ $m^3$) | Scenario of Supply warranty for SPRM | Supply warranty time percentage (%)* |
|---|---|---|---|---|
| (0, 31) | 0 | 3.38 | 100% water availability | 1 |
| (0, 90) | 6 | 3.58 | 100% water availability | 0.34 |
| (0, 180) | 10 | 3.71 | Water availability with storage dependency | 0.17 |
| > 365 | 17 | 3.95 | Water deficit (multi-year droughts) | 0.084 |

* **As [100% Supply warranty time during 31 days / Analysis Scenario of Supply warranty time (days)]**

**Table 3.** The Cantareira Equivalent System (ES) performance criteria for Calibration-Validation periods. *Cal.
=Calibration period and Val. =Validation period. The calibration and validation performance criteria for each
basin in the system can be found in the "Complementary Material" - supplementary material - section A. – Table
A-1.

| Cantareira Equivalent System | Area ($km^2$) | VE | | NSE | | $NSE_{Log}$ | | RSR | | $R^2$ | | PBIAS (%) | |
|---|---|---|---|---|---|---|---|---|---|---|---|---|---|
| | | Cal. | Val. | Cal. | Val. | Cal. | Val. | Cal. | Val. | Cal. | Val. | Cal. | Val. |
| | 2265.0 | 0.91 | 0.80 | 0.95 | 0.90 | 0.94 | 0.74 | 0.21 | 0.38 | 0.96 | 0.92 | -3.40 | -12.36 |













**Table 4.** Economic profit loss projection scenario for the period 2007-2040 (x$10^6$ US$)

| RCM scenario | RCP scenario | Demand scenario | 2007-2040 | | | | | |
|---|---|---|---|---|---|---|---|---|
| | | | $Rp_2$ | Dif.% | $Rp_{10}$ | Dif.% | $Rp_{100}$ | Dif.% |
| Eta-MIROC5 | 4.5 | SD | 13818 | 17.13 | 19696 | 27.18 | 22965 | 32.61 |
| | | NSD | 16674 | | 27049 | | 34079 | |
| | 8.5 | SD | 19953 | 16.73 | 28443 | 26.82 | 33035 | 32.54 |
| | | NSD | 23961 | | 38865 | | 48971 | |
| Eta-HADGEM | 4.5 | SD | 14713 | 8.80 | 25254 | 13.36 | 32242 | 14.61 |
| | | NSD | 16132 | | 29146 | | 37758 | |
| | 8.5 | SD | 13667 | 8.62 | 23440 | 13.15 | 29761 | 14.54 |
| | | NSD | 14956 | | 26990 | | 34825 | |

**Note: SD: stationary and NSD: non-stationarity**

**Table 5.** Economic profit loss projection scenario for the period 2041-2070 (x$10^6$ US$)

| RCM scenario | RCP scenario | Demand scenario | 2041-2070 | | | | | |
|---|---|---|---|---|---|---|---|---|
| | | | $Rp_2$ | Dif.% | $Rp_{10}$ | Dif.% | $Rp_{100}$ | Dif.% |
| Eta-MIROC5 | 4.5 | SD | 10168 | 50.28 | 14487 | 56.34 | 16788 | 59.84 |
| | | NSD | 20453 | | 33178 | | 41799 | |
| | 8.5 | SD | 8733 | 61.61 | 12498 | 66.09 | 14378 | 69.06 |
| | | NSD | 22747 | | 36855 | | 46476 | |
| Eta-HADGEM | 4.5 | SD | 10232 | 30.44 | 17550 | 33.91 | 22316 | 34.98 |
| | | NSD | 14710 | | 26555 | | 34321 | |
| | 8.5 | SD | 8544 | 36.24 | 14645 | 39.41 | 18594 | 40.26 |
| | | NSD | 13399 | | 24170 | | 31125 | |

**Note: SD: stationary and NSD: non-stationarity**

**Table 6.** Economic profit loss projection scenario for the period 2007-2040 (x$10^6$ US$)

| RCM scenario | RCP scenario | Demand scenario | 2071-2099 | | | | | |
|---|---|---|---|---|---|---|---|---|
| | | | $Rp_2$ | Dif.% | $Rp_{10}$ | Dif.% | $Rp_{100}$ | Dif.% |
| Eta-MIROC5 | 4.5 | SD | 14698 | 53.45 | 20956 | 59.20 | 24237 | 62.47 |
| | | NSD | 31575 | | 51367 | | 64582 | |
| | 8.5 | SD | 7929 | 60.23 | 11338 | 64.93 | 13017 | 68.04 |
| | | NSD | 19938 | | 32332 | | 40734 | |
| Eta-HADGEM | 4.5 | SD | 8508 | 49.19 | 14569 | 51.80 | 18459 | 52.81 |
| | | NSD | 16743 | | 30225 | | 39116 | |
| | 8.5 | SD | 16553 | 22.40 | 28392 | 26.31 | 36213 | 27.39 |
| | | NSD | 21329 | | 38532 | | 49873 | |

**Note: SD: stationary and NSD: non-stationarity**







**Figures.**

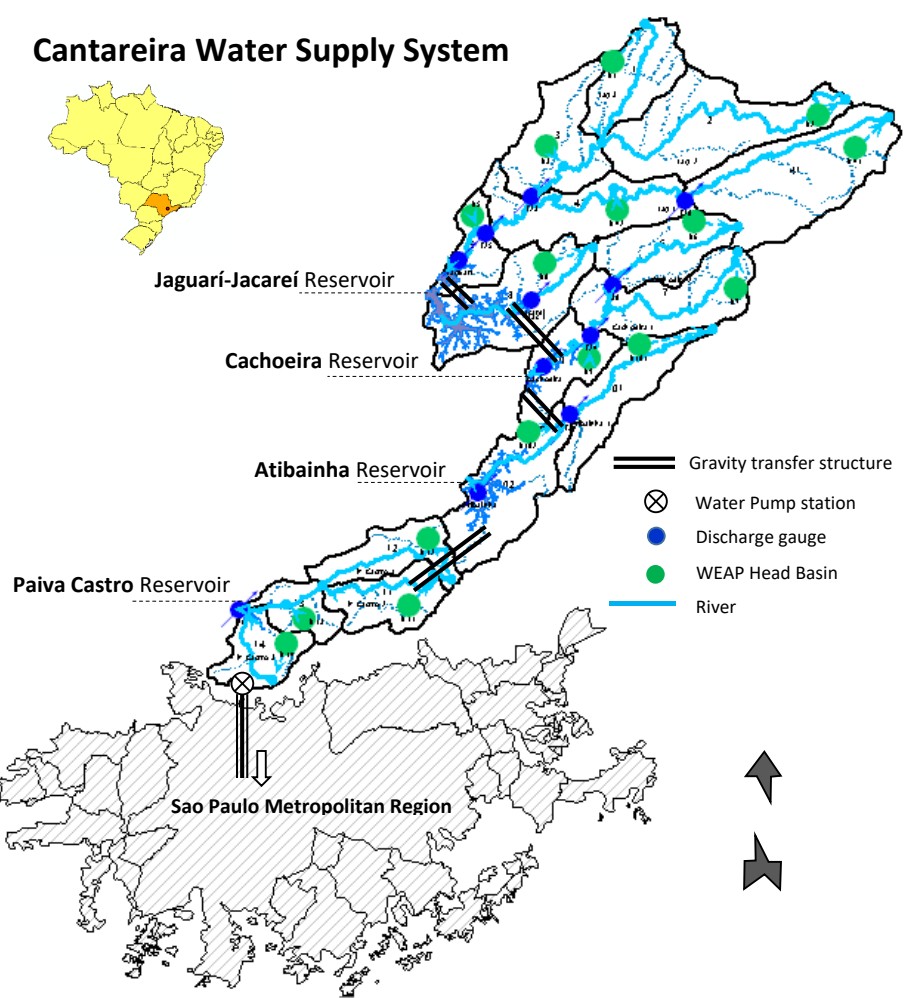


**Figure 1.** System structure composition and catchment areas: Jaguarí-Jacareí, Cachoeira, Atibainha and Paiva
Castro watersheds.





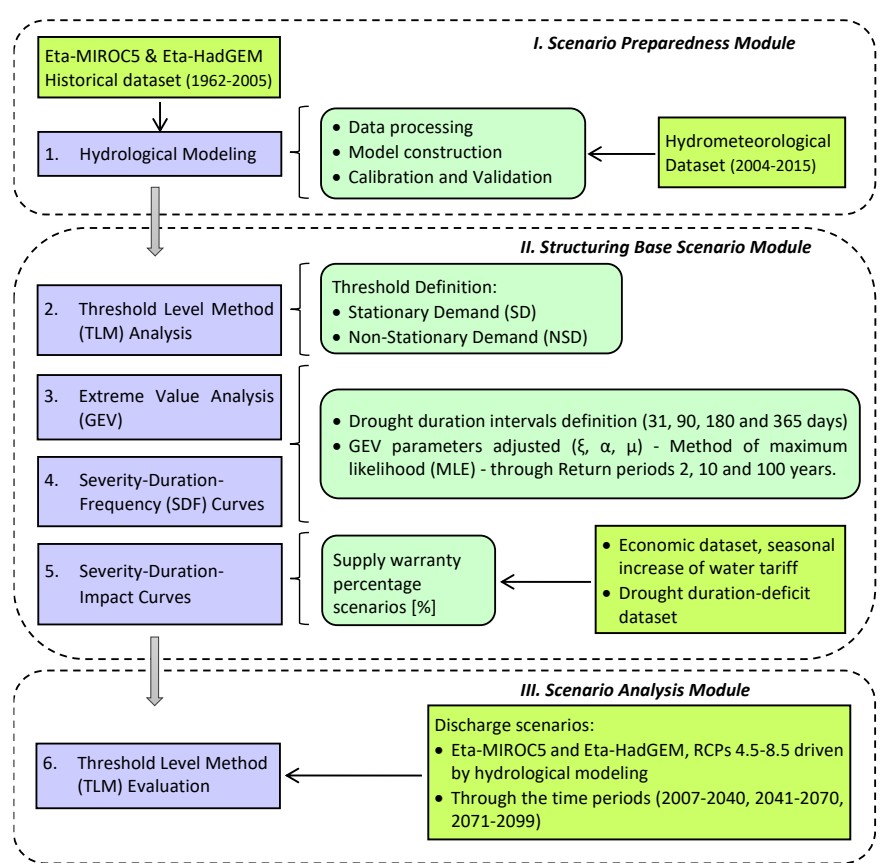


**Figure 2.** Methodology flowchart and main inputs.











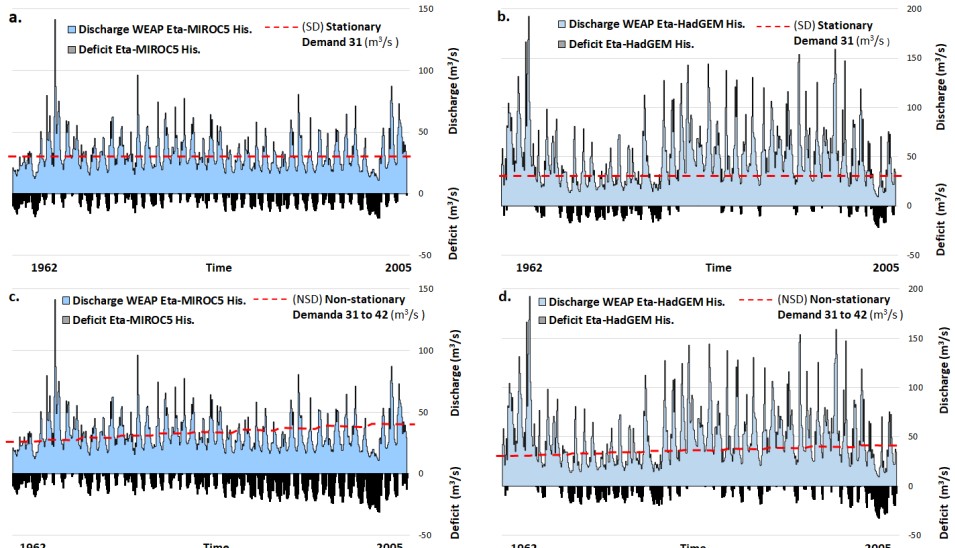


**Figure 3.** TLM Evaluation from historical discharge WEAP-Eta scenarios, under Stationary (SD) and Non-

Stationary Demand (NSD) assumptions as the "threshold level": a. 31 m³/s and Eta-MIROC5. b. 31 m³/s and

Eta-HadGEM. c. 31 to 42 m³/s and Eta-MIROC5. d. 31 to 42 m³/s and Eta-HadGEM.


















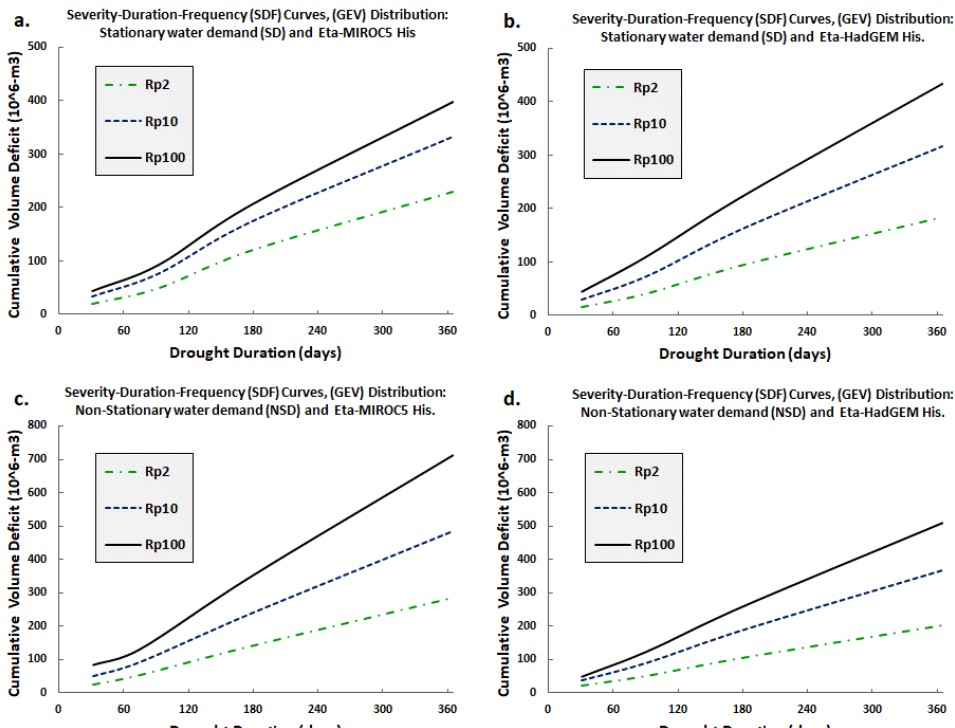


**Figure 4.** SDF curves under stationary and non-stationary demand assumptions and historical discharge WEAP-
Eta scenarios: a. (SD) 31 m$^3$/s and Eta-MIROC5. b. (SD) 31 m$^3$/s and Eta-HadGEM. c. (NSD) 31 to 42 m$^3$/s and
Eta-MIROC5. d. (NSD) 31 to 42 m$^3$/s and Eta-HadGEM.











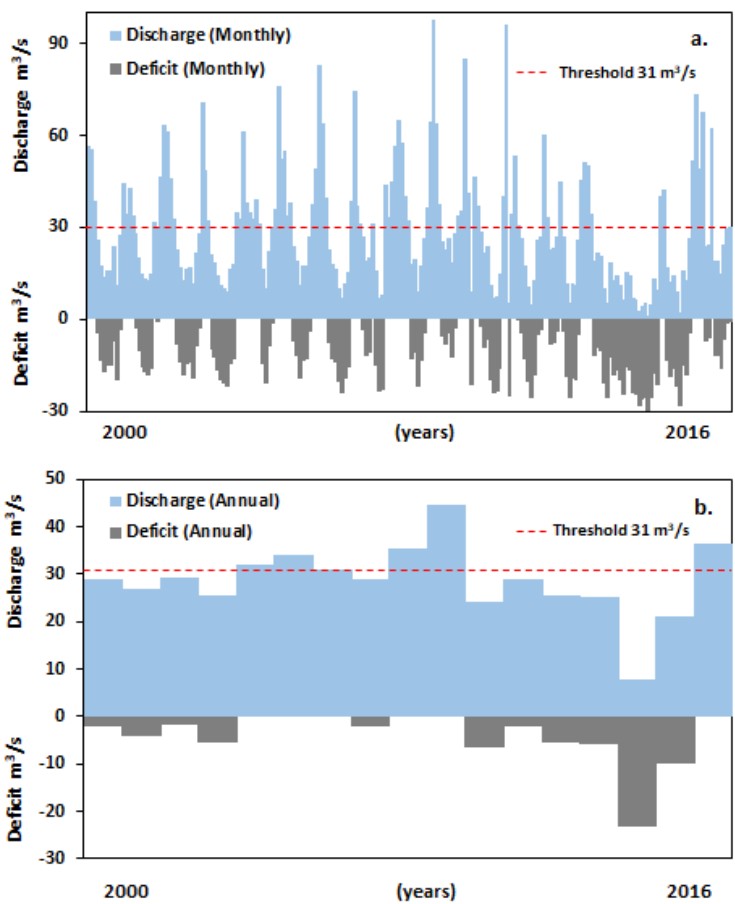


**Figure 5.** TLM analysis under two discharge scenarios, 2000-2016 period. a) Monthly average discharge and b)

Annual average discharge.












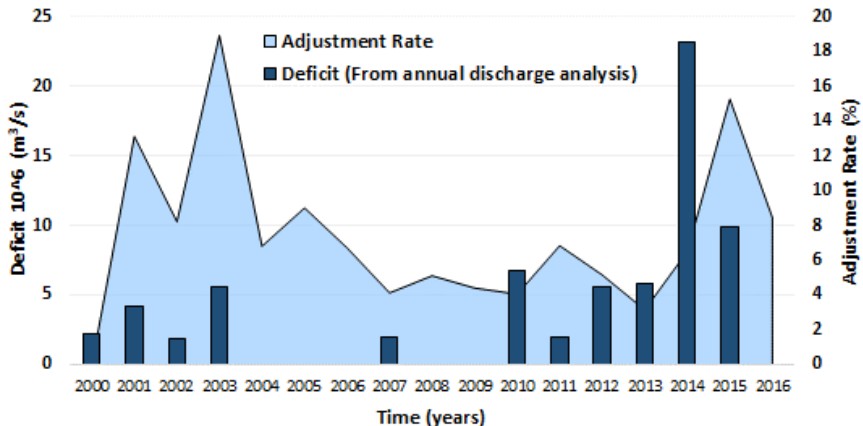

**Figure 6.** Co-evolution of the drought deficit and price adjustment rates (SABESP – Cantareira System) during
2000-2016 period. Note: deficits defined from TLM analysis under a demand threshold of 31 m3/s and annual
average discharge

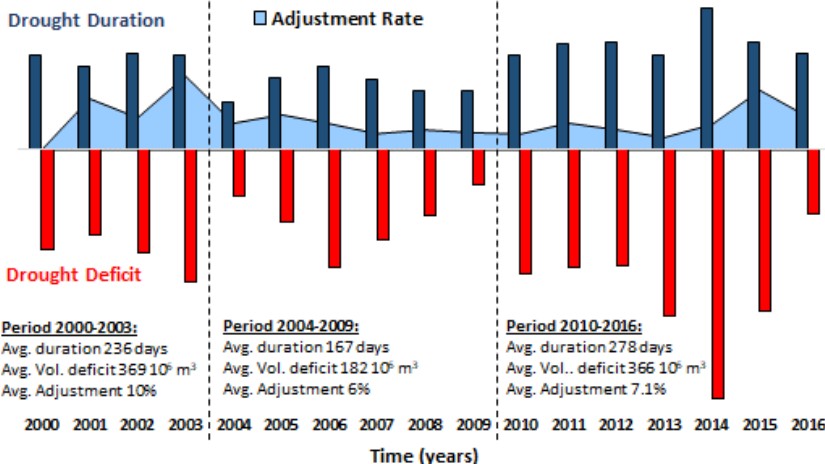


**Figure 7.** Empirical relationship between Cantareira System drought duration "blue-bar in days" [derived from
monthly average discharge analysis], Cantareira System drought deficit "red-bar in $10^6$-m$^3$" [assessed from
monthly average discharge analysis] and annual price adjustment rates under variate hydrological conditions in
percentage.






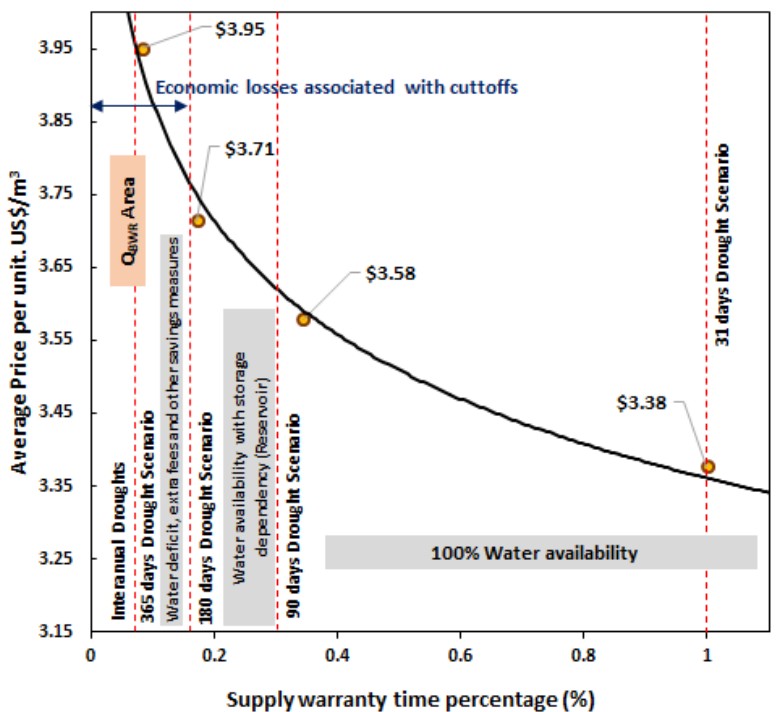


**Figure 8.** Cantareira System demand curve based on the supply warranty time percentage

















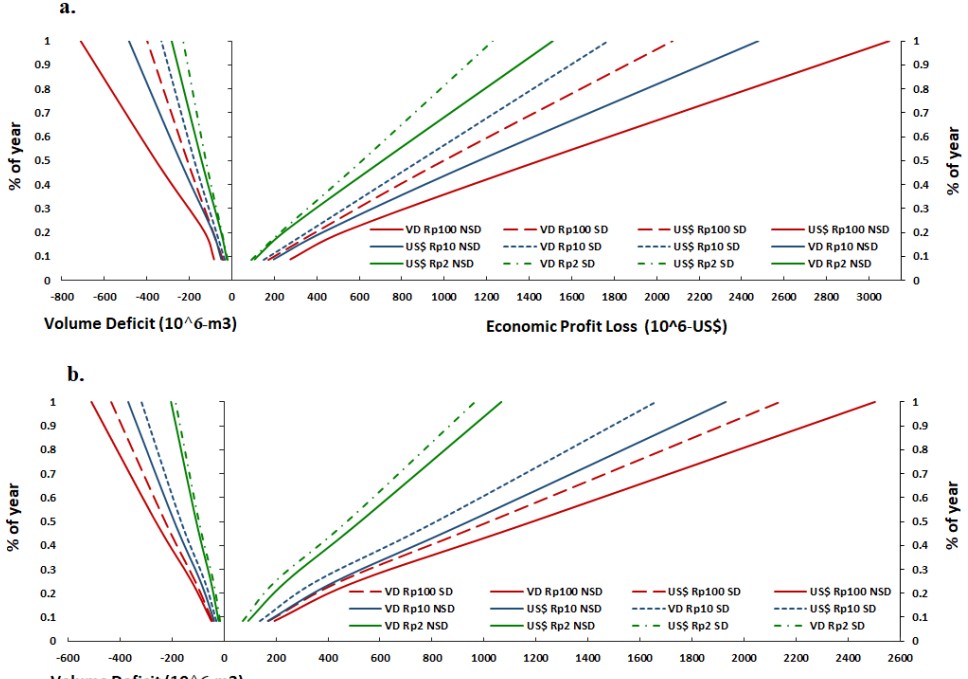

**Figure 9.** Severity-Duration-Impact curves. Sector **a.** Severity-Duration-Frequency-Profit Loss under the historical *Eta-MIROC5* scenario. Sector **b.** Severity-Duration-Frequency-Profit Loss under the historical *Eta-HadGEM* scenario. Note: *SD* and *NSD* are the stationary or non-stationary demands, respectively; "*VD*" is the volume deficit, under return period of 2, 10 and 100 years; *% of year* is the drought event duration in relation to one year.





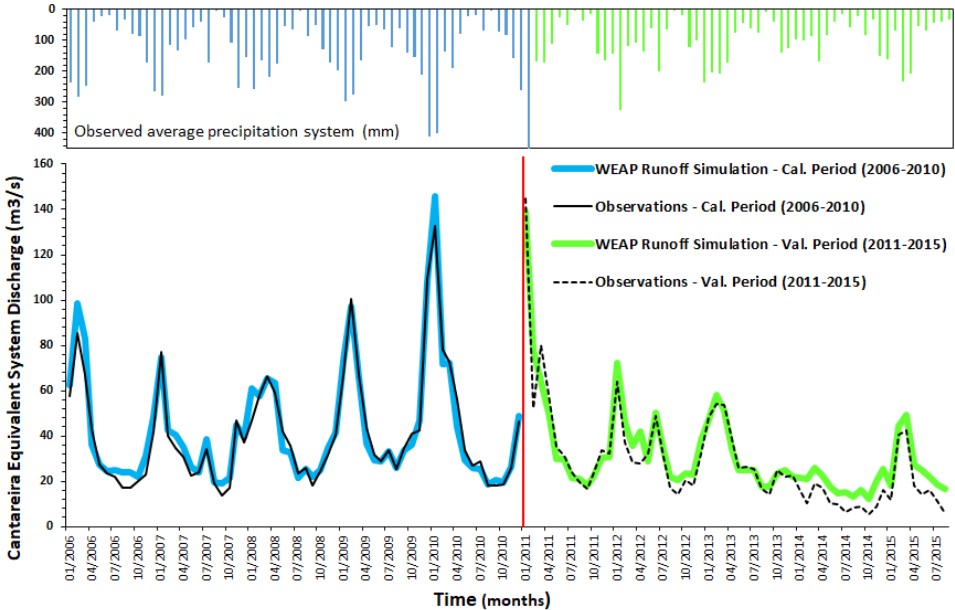

992

**Figure 10.** WEAP Hydrographs, Calibration period (2006-2010) and Validation period (2011-2015)

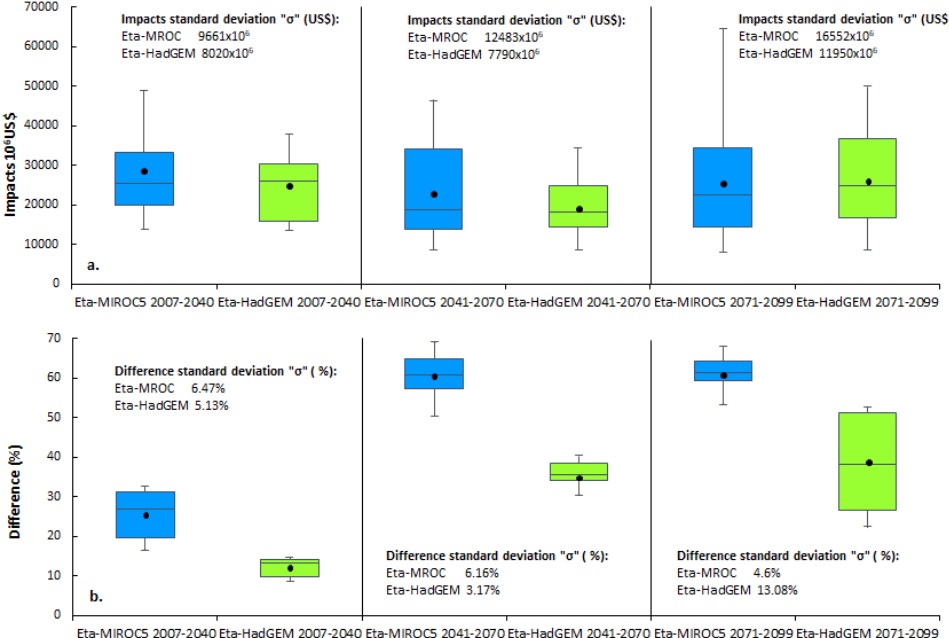

994

**Figure 11.** Box plots with impacts and relative differences between climate change scenarios. Sector a:
Economic impacts under time periods of climate change scenarios. Sector b: Percentage difference between the
demand scenarios under time periods of climate change scenarios.