# Peer review of "Economic impacts of drought risks for water utilities through"

_Hydrology and Earth System Sciences, 2017_

## Author Comment (AC1) · 22 Nov 2017

Author's typing error.

Correct reference: Guzmán, D. A.

Diego Alejandro Guzmán.

---

## Referee Comment (RC1) · Anonymous Referee #1 · 11 Mar 2018

This manuscript aims to develop a method to estimate future economic impacts of climate change on water utilities. Specifically, it uses severity-duration-frequency analysis of rainfall projected by a regional climate model to assess financial risk to the Sao Paolo metropolitan water supply.

I found this the most challenging manuscript I have review in quite a while. On the one hand, there is a great need for studies finding ways to translate hydrometeorological predictions into impact assessments, and this study tries to do just that. On the other hand, in such a study one still looks for some generic insights that advance the field.

[Figure]

This is where I struggled.

I do believe the conventional standard of scientific evidence can be relaxed a bit to in the case of inter-disciplinary papers of a more applied nature, which try to make science relevant to socio-economic decisions in the presence of uncertainty. However such interdisciplinary papers have requirements of their own, part of which derive from the fact that no reader is likely to be an expert in all areas of the methodology and that large uncertainties occur throughout the analysis chain.

To address this, the authors should provide more detailed discussion on:

1) Discussion of issues or uncertainties in pre-existing methods and techniques used, both those that are known (upfront, in the methods) and those that are newly found in interpretation (in the discussion).

2) Justification of any new methods, techniques or assumptions that are made.

3) In the discussion; the overall most tenuous assumptions, and therefore ultimately the greatest caveats and uncertainties, that need to be considered when basing practical decisions on this study.

4) A research agenda that provides new insights into what specifically would need to happen to make studies such as this more suitable for decision makers.

From the hydrological perspective, the methods are not novel and open to varying degrees of criticism, but broadly competent and probably acceptable for this type of analysis, although that still needs to be argued. I have some doubts about the assumptions made in the economical analysis, but I have very little expertise in economic analysis and I recommend the editor seek an additional review from at least one economist.

At face value, I found it hard to believe that water utility profit loss could realistically reduce regional GDP by as much as 10%. If that were the case, than would that not imply that the utility is one of the main employers? Presumably that level of loss would well exceed the company's capacity, sending it bankrupt well before that, leaving the gov-
ernment to deal with the fallout, and moving the scenario beyond your assumptions? As I said I am not an economist, but something seems not quite right there.

The English is generally very readable but a few issues occur more than once:

* Long sentences with unnecessary clauses (e.g., p1, l15-17). Please try to shorten and simplify such sentences without using clauses where possible.

* Poor word choice. For example 'to prioritize' (p1,l11) is a mental activity by (in this case) people, and cannot be done by phenomena. Also, do 'establishments' (p3,l91) relate to households or businesses?

* Inconsistent form (single/plural) between noun and verb (e.g., p1, l20)

* Incorrect combination of verb and preposition (e.g. p1,l31 should be "associated with")

Most of these would likely be picked up in proofreading by a native English speaker.

---

## Referee Comment (RC2) · Anonymous Referee #2 · 17 May 2018

The manuscript entitled 'Economic impacts of drought risks for water utilities [assessed] through SDF framework under climate change scenarios, presents a case study on a timely topic. The study combines hydrological modelling of future scenarios with drought frequency analysis tied to a hydrologic-economic risk assessment for a very specific and certainly interesting case. The authors claim they aim to 'describe an academic exercise to manage drought financial planning...' (p.4). Similar to Reviewer-1 my main criticism of the manuscript also relates to the lack of clarity on the wider generalized contribution of the study. Despite the plural 'water utilities...' in the title and

the phrased aim of an 'academic exercise', the manuscript reads mostly like a report on a case-specific study. As a reader I struggled to extract what was previously known and available and what is the main new contribution of the study and how the knowledge gained can inform further progress. I will provide some details below. In its current form, the pure case study descriptions is too long and the transferable part too small. In particular an extensive discussion and debate on assumptions, uncertainties and influences on applicability, which will make this useful to an international readership is missing. While I think the study has potential, I am afraid that with this balance of information given, unfortunately I cannot recommend publication in HESS.

The manuscript will also require a number of clarifications and improvements to structure. These may include:

- clarifying the actual academic objectives and/or hypotheses

- cutting and cleaning of a lot of unnecessary local and specific information that are not relevant or at least take attention away from the envisioned 'academic exercise' (such as e.g. lines 134-137, but many more also)

- a clear separation of a generic concise methods section from the prior assumptions and also from the specific results of this study. The description of study area and methods is nine pages, followed by only three and a half pages of results and conclusions. A discussion putting the results into the wider context is missing. This imbalance illustrates clearly that the commonly accepted structure of a science paper is not followed.

- In the current manuscript, most figures are already referred to in the methods section, which then contains already many details on results and is consequently rather confusing. To the reader it is unclear, which exactly are the new results reported here vs material available prior to this contribution.

- line 91 - why 'are affected' when referring to past droughts? Otherwise this is water scarcity and has perhaps not to do with drought as a temporary climate phenomenon.

This is also something I struggled with - clear definitions would be very helpful to the reader.

- The threshold appear not close to usually employed drought thresholds which often represent the 10th or 20th percentile of the empirical distribution function of river flow or other hydrological flux or state variables. Where does this demand threshold rank in the flow duration curve? And can the deficit then really be termed drought. Some of the figures indicate a deficit every year. Commonly this may not be consisidered a drought (as an unual and rare event). A thorough discussion and explanation comparing to the literature is needed on this aspect.

- The WEAP modelling is presented as part of the study, but there is not enough information on model details to convince the reader of a carefully carried out modelling. It also lacks an assessment of uncertainty. In a climate change application the most important information is whether the signal of change exceeds that of current uncertainty. Some of the modelling description uses unusual terminology and phrasing. I suggest to have this proofread by a modelling expert.

Specific comments

75 - sentence makes no sense grammatically

156 - text unassigned to a header and structurally unclear

91 - what are 'establishments' ?

396-397 unnecessary

Table 1 - Those are not 'variables' - wrong terminology Table 2 - should it not be 30, 90 and 90,180? Table 3 - First two columns don't make sense in the table as they don't vary Tables 4-6 would be much nicer as graphs - this illustrate the point: abolute numbers don't matter to an international readership that will not be ingterested in the specific case, but in sensitivies, systematic differences, trends etc...

[Figure]

Figure 1 - are there numbers in the subwatersheds? resolution is insufficient - either show clearly or remove from map

Figure 2 - Why show the seasonality if what matters is the demand vs the deficit over the year?

Overall there many abbreviations that sometimes make reading difficult.

---

## Author Comment (AC2) · 14 Jun 2018

Dear editor and reviewers,

We thank you very much for the valuable comments and suggestions about our manuscript "Economic impacts of drought risks for water utilities through Severity-Duration-Frequency framework under climate change scenarios" in HESSD (Hydrology and Earth System Science Discussion). We performed a careful revision to make all suggested changes and we believe the manuscript is now much improved. Please

check if you are happy with the new version and let us know if you have any further questions or additional suggestions. You will find in blue the responses to each comment below. All changes to comply with the reviewers' suggestions were highlighted in yellow in the manuscript. Yours sincerely,

Diego Alejandro Guzman Arias Corresponding author

Anonymous Referee #1 () Provide more detailed discussion on:

1) Discussion of issues or uncertainties in pre-existing methods and techniques used, both those that are known (upfront, in the methods) and those that are newly found in interpretation (in the discussion).

In the restructuring of the document, a sub-chapter was included in Results and discussions "4.4 Considerations on Uncertainties". Lines 563-595.

2) Justification of any new methods, techniques or assumptions that are made.

In the chapter "3.3 Water price and Hydrological drought relationship" the main assumptions of the method were established, all the changes and new stretches are highlighted in yellow.

3) In the discussion; the overall most tenuous assumptions, and therefore ultimately the greatest caveats and uncertainties, that need to be considered when basing practical decisions on this study.

The new chapter "4.4.Considerations on Uncertainties" addresses the main sources of uncertainty in our work flow, shortly the uncertainties in the modeling chain (climate-hydrology) and the socio-economic considerations (water demand, water tariff policies, water supply system operation). The uncertainties of the modeling chain have been extensively addressed in the scientific literature, and we gathered main points and useful references; whilst the considerations on the study case, i.e. on economy factors and the water supply system are less general. As mentioned in the chapter, we considered

the current reservoir operation, water tariff policy and state water use policy, all of which affect the SDF curves and also the pricing strategy during a drought event. Therefore applications with same objective, even in different cases would face these uncertainties. With that information in hand, or prospected ones, however, the methodology is versatile to respond to these configurations. Additionally, our considerations added in the Conclusions and recommendations, lines 665-660 support the usefulness of this methodology for planning in the long-term, but our developments did not consider long multi-year droughts, in which diverse less secure strategies could take place during the event, for example implementing quickly, at a great cost, and less robustly alternative water sources, which fall outside our considerations for supply and would bring further consequences on water tariff strategies.

4) A research agenda that provides new insights into what specifically would need to happen to make studies such as this more suitable for decision makers. Lines 646-654 in conclusions and recommendations, a work agenda for future research is proposed.

At face value, I found it hard to believe that water utility profit loss could realistically reduce regional GDP by as much as 10%. If that were the case, than would that not imply that the utility is one of the main employers? Presumably that level of loss would well exceed the company's capacity, sending it bankrupt well before that, leaving the government to deal with the fallout, and moving the scenario beyond your assumptions? As I said I am not an economist, but something seems not quite right there. We appreciate the comment, this was perhaps a big typing error (millions for billions). On re-structuring text, the paragraph was removed from the abstract, corrected and relocated to results. This was done, seeking to give a more general and less regional connotation to the results as suggested by the reviewer (lines 495 to 502).

The English is generally very readable but a few issues occur more than once: The document was reviewed by a person with English as their native language.

* Long sentences with unnecessary clauses (e.g., p1, l15-17). Please try to shorten

and simplify such sentences without using clauses where possible.

Modified text within the body of the abstract (p1, l13 to l35) and in other sections of the text through the grammatical revision.

* Poor word choice. For example, 'to prioritize' (p1, l11) is a mental activity by (in this case) people, and cannot be done by phenomena. Also, do 'establishments' (p3, l91) relate to households or businesses?

Modified text within the body of the abstract (p1, l13 to l35) and in other sections of the text through the grammatical revision. About the word "establishments" this refers to the economic impacts on several sectors that depend of water for its operation, as can be households, business and industrial; this can be verified in Marengo et al., 2015 "A seca e a crise hídrica de 2014-2015 em São Paulo". In the text it was clarified (p3, l84).

* Inconsistent form (single/plural) between noun and verb (e.g., p1, l20)

Modified text within the body of the abstract (p1, l13 to l35) and in other sections of the text through the grammatical revision.

* Incorrect combination of verb and preposition (e.g. p1, l31 should be "associated with")

Modified text within the body of the abstract (p1, l13 to l35)

Most of these would likely be picked up in proofreading by a native English speaker.

Please also note the supplement to this comment:
https://www.hydrol-earth-syst-sci-discuss.net/hess-2017-615/hess-2017-615-AC2-supplement.pdf

---

## Author Comment (AC3) · 14 Jun 2018

Dear editor and reviewers,

We thank you very much for the valuable comments and suggestions about our manuscript "Economic impacts of drought risks for water utilities through Severity-Duration-Frequency framework under climate change scenarios" in HESSD (Hydrology and Earth System Science Discussion). We performed a careful revision to make all suggested changes and we believe the manuscript is now much improved. Please

check if you are happy with the new version and let us know if you have any further questions or additional suggestions. You will find in blue the responses to each comment below. All changes to comply with the reviewers' suggestions were highlighted in yellow in the manuscript. Yours sincerely,

Diego Alejandro Guzman Arias Corresponding author

Anonymous Referee #2 ()

As a reader I struggled to extract what was previously known and available and what is the main new contribution of the study and how the knowledge gained can inform further progress. I will provide some details below. In its current form, the pure case study descriptions are too long and the transferable part too small.

The manuscript will also require a number of clarifications and improvements to structure. These may include:

The comments of the reviewer from number 1 to 6 and 8, emphasize the structure and clarifications that were accepted by us in the new manuscript.

1. Clarifying the actual academic objectives and/or hypotheses.

2. Cutting and cleaning of a lot of unnecessary local and specific information that are not relevant or at least take attention away from the envisioned 'academic exercise' (such as e.g. lines 134-137, but many more also)

3. A clear separation of a generic concise methods section from the prior assumptions and also from the specific results of this study. The description of study area and methods is nine pages, followed by only three and a half pages of results and conclusions. A discussion putting the results into the wider context is missing. This imbalance illustrates clearly that the commonly accepted structure of a science paper is not followed.

4. In the current manuscript, most figures are already referred to in the methods section, which then contains already many details on results and is consequently rather confusing. To the reader it is unclear, which exactly are the new results reported here vs material available prior to this contribution.

5. Line 91 - why 'are affected' when referring to past droughts? Otherwise this is water scarcity and has perhaps not to do with drought as a temporary climate phenomenon.

6. This is also something I struggled with - clear definitions would be very helpful to the reader.

7. The threshold appears not close to usually employed drought thresholds which often represent the 10th or 20th percentile of the empirical distribution function of river flow or other hydrological flux or state variables. Where does this demand threshold rank in the flow duration curve? And can the deficit then really be termed drought. Some of the figures indicate a deficit every year. Commonly this may not be considered a drought (as an annual and rare event). A thorough discussion and explanation comparing to the literature is needed on this aspect.

We appreciate the comments. Regarding the threshold, some authors do not restrict the use to indexes or variables of the state of the flow, for example some value of the duration curve. Hisdal et al., 2004 and JH Sung and Chung, 2014, define the threshold from "thresholds of desired performance", in our case the threshold is pre-established by the characteristics of water withdrawal for the SPMR, condition on which you want to evaluate the system. On the other hand, during the introduction the clarification was made about the characteristics of the hydrological drought and what leads to the water deficit, our main focus. This assumption of annual deficits, is clearly an assumption for the implementation of the method, however, during the stage of the calculation of the impact, minor droughts are not considered.

8. The WEAP modelling is presented as part of the study, but there is not enough information on model details to convince the reader of a carefully carried out modelling. It also lacks an assessment of uncertainty. In a climate change application, the most important information is whether the signal of change exceeds that of current uncertainty. Some of the modelling description uses unusual terminology and phrasing. I suggest to have this proofread by a modelling expert.

Specific comments:

- Sentence makes no sense grammatically Modified text within the body of the abstract (p1, l13 to l35) and in other sections of the text through the grammatical revision.

- Text unassigned to a header and structurally unclear The text was complemented and organized according to the request of the reviewer.

- What are 'establishments'? The term was revised and replaced in lines 82 to 85 of the new text

396-397 unnecessary The text was complemented and organized according to the request of the reviewer.

Table 1 - Those are not 'variables' - wrong terminology Table 2 - should it not be 30, 90 and 90,180? The table was modified.

Table 3 - First two columns don't make sense in the table as they don't vary The table was introduced in the new Figure 6 and the columns were eliminated.

Tables 4-6 would be much nicer as graphs - this illustrate the point: absolute numbers don't matter to an international readership that will not be interested in the specific case, but in sensitivities, systematic differences, trends etc... These tables were changed by Figures 9, 10 and 11 in the manuscript.

Figure 1 - are there numbers in the sub-watersheds? Resolution is insufficient – either show clearly or remove from map Figure 1 was edited and new descriptors were added for better understanding.

Figure 2 - Why show the seasonality if what matters is the demand vs the deficit over the year? We did not find this description in Figure 2 of what was commented by the

[Figure]

reviewer.

Please also note the supplement to this comment:
https://www.hydrol-earth-syst-sci-discuss.net/hess-2017-615/hess-2017-615-AC3-supplement.pdf

[Figure]

**Supplement:**

**Economic impacts of drought risks for water utilities through Severity-Duration-Frequency assessment under climate change scenarios**

Diego A. Guzman[1], Guilherme S. Mohor[2], Denise Taffarello[3] and Eduardo M. Mendiondo[3]

[One] Department of Civil Engineering, Pontificia Bolivariana University, Bucaramanga, STD, 681007, Colombia

[Two] Institute of Earth and Environmental Science, University of Potsdam, Karl-Liebknecht-Str. 24–25, 14476 Potsdam, Germany

[3] Department of Hydraulics and Sanitation - Sao Carlos School of Engineering, University of Sao Paulo - Sao Carlos, SP, 13566-590, Brazil

*Correspondence to*: Diego A. Guzman (daga2040@hotmail.com)

**Abstract**

Climate change and increasing water demands show the need for implementing planning strategies for financial sustainability of sectors, which are highly dependent on water resources, such as water utility companies. The financial vulnerability of these companies increases considering water supply growth and low availability scenarios, resulting in less profits or economic bankrupt. Generally, methods to estimate financial impacts caused by drought are not as numerous and clear as those for floods due to the complex characteristics of the phenomenon. Therefore, we propose a new assessment to estimate the business interruption cost considering the uncertainties in the climate and urban demand projections in the medium and long term. The methodology integrates the semi-distributed hydrological simulation procedures linked to the Water Evaluation and Planning system (WEAP) under radiative climate forcing scenarios RCP 4.5 and 8.5 from the regional climate model outputs Eta-INPE/MIROC5 and HadGEM-ES (RCM). The approach continues with the hydrological drought assessment "Severity-Duration-Frequency" (SDF), based on stationary and non-stationary water demand assumptions to establish the method's threshold levels. Likewise, the methodology defines a water tariff price delimited by the drought duration and the system's robustness analysis to determine revenue loss scenarios in the water utility, through planning periods: 2007-2040, 2041-2070, and 2071-2099. As a case study, the approach is applied to the Cantareira Water Supply System in the São Paulo Metropolitan Region (SPMR), the main water supply source for about 11 million people. The results show that the water-cost outputs based on Eta-MIROC5 present higher revenue losses in the company than those based on HadGEM-ES. Meanwhile, the relationship between RCP scenarios 4.5 and 8.5 showed lower variability compared to the analyzed climate-with-water demand scenarios. However, the Non-stationary demand (NSD)

trend imposed larger differences in the drought resilience financial gap, suggesting that the demand-related uncertainty would be far greater than that associated with climate sensitivity.

**Key Words:** Climate change, Severity-Duration-Frequency assessment, Water utility revenue losses, Hydrological droughts

**1. Introduction**

Climate change, population growth and uncontrolled urban/industrial development make society more dependent on water (Montanari et al., 2013). The complex interaction between meteorological, terrestrial and socio-economic water distribution schemes are the main factors that define droughts (Lloyd-hughes, 2013; Van Loon et al., 2016b, 2016a; Wada et al., 2013). Therefore, in order to address a potential drought scenario in the future with demand as a determinant anthropogenic factor, society is required to rethink the way forward, mainly to reduce its vulnerability by regulating its demand (Falkenmark and Lannerstad, 2004; Kunreuther et al., 2013; Wanders and Wada, 2015).

Apparently, droughts may not be as apparent as floods, but have proven to be one of the most complex risks due to their slow development, strong and long lasting impacts as well as broad geographic coverage (Bressers and Bressers, 2016; Frick et al., 1990a; Smakhtin and Schipper, 2008; Van Lanen et al., 2013). Furthermore, various studies have shown that more severe and prolonged droughts are expected for the future, leading to greater economic consequences, environmental degradation and loss of human lives (Asadieh and Krakauer, 2017; Balbus, 2017; Berman et al., 2013; Freire-González et al., 2017; Prudhomme et al., 2014; Shi et al., 2015; Stahl et al., 2016; Touma et al., 2015). Therefore, it is essential to create adequate risk perception, aiming to reduce the risks, mitigate the impacts and build a more resilient-drought community (Bachmair et al., 2016; Mishra and Singh, 2010; Nam et al., 2015).

The most visible impacts on the urban water supply are strongly related to hydrological droughts and not directly to meteorological droughts (Bachmair et al., 2016; Van Lanen et al., 2016). A hydrological drought is defined as a negative anomaly in surface and subsurface water levels (Mishra and Singh, 2010; Van Loon, 2015; Wanders et al., 2017). These negative anomalies on the surface, related to an excessive level of water demand can cause water systems to collapse (Mehran et al., 2015; Van Loon et al., 2016b; Wanders and Wada, 2015). Therefore, in this study we address hydrological droughts as the main driver of business interruption in the water utility company, specifically when urban water demand exceeds the supply system offer (Bressers and Bressers, 2016; Frick et al., 1990a, 1990b).

The definitions of drought losses (or drought costs) are not as clear as those regarding floods or methods for estimating drought costs, although diverse, not as numerous as floods. (Freire-

González et al., 2017; Logar and van den Bergh, 2013; Meyer et al., 2013). In a comprehensive review by Logar and van der Bergh (2012), the authors suggest a division of drought costs as direct, indirect and non-market costs. Furthermore, Meyer et al. (2013) suggest extra categories, differentiating Business interruption costs as primary tangible costs, although not configured as

"due to direct physical contact". Despite a diversified range of methods presented by Meyer et al. (2013) and Logar and van der Bergh (2012), several are either: for non-tangible or indirect methods, specific for the agricultural sector or economy wide oriented (i.e. fit to a broader scale application) which in our case would likely incur in less precise results. Regarding the allocation of water companies by reduced water availability, several approaches seem to be adequate, such as market valuation techniques or ex-post evaluations, that is, comparing changes in GDP or changes in price between affected and unaffected years

In Brazil, from 2013 to 2015, the population of the Sao Paulo Metropolitan Region (SPMR)

experienced the most acute water crisis in its history (Coutinho et al., 2015; Nobre and

Marengo, 2016; Taffarello et al., 2016). According to the Federation of Industries of the State of Sao Paulo (FIESP), it was estimated that 60,000 households, business and industrial sectors, which represent almost 60% of the state's industrial GDP, were affected by a lack of water (Marengo et al., 2015). Likewise, during 2014 and 2015, the Sao Paulo State Water Utility

Company (SABESP) recorded an average annual liquid net income reduction of approximately

63% compared to 2013, leading to a major financial crisis in the company (GESP, 2016;

SABESP, 2017a). To analyze the water utility drought impacts, several control strategies are usually implemented as price-based policies that seek to change the user's consumption pattern based on economic penalties or incentives (Buurman et al., 2017; Millerd, 1984; Rossi and

Cancelliere, 2013). However, the implementation of these strategies entails a great complexity in their planning and high risks of economic impacts for the water company (SABESP, 2015;

Watts et al., 2012).

To deal with global change, understanding the interplay between multiple drivers of risks and socioeconomic development is increasingly required to inform effective actions to manage new drought risks and pursue sustainable development. However, as long as there are no systematic and detailed studies on the assessment drought impacts on the regional economy, shaping financial planning policies is a complex and uncertain task that must be reinforced. Therefore, based on the drought Severity-Duration-Frequency characterization, we explore the water utility company business interruption cost assessment by integrating an analysis framework driven by climate change, water-demand scenarios and the supply system robustness. This paper describes an academic exercise to manage drought financial planning, running the Eta-INPE (RCM) outputs, through a semi-distributed hydrological model of the water supply system developed using WEAP.

The sections of this article outline interconnected methods and criteria, explained as follows. In Section 2, the text describes the study area and water crisis contextualization. Section 3 outlines the methodological approach starting with the hydrological modeling, characterization of the droughts using the threshold level method, the formulation of the SDF curves of the system and subsequently, the climatic, hydrological and economic aspects of the methodology. In Section 4, the results and discussions are shown as financial drought planning scenarios. Finally, in Section 5, the conclusions and recommendations are presented regarding the proposed approach.

**2. Study area and water crisis contextualization.**

The Cantareira Water Supply System, hereafter referred to as the Cantareira System, is located in South-East Brazil between the states of Sao Paulo and Minas Gerais. The rainy season in the Cantareira System generally begins at the end of September and ends in March. In this period, on average 72% of the rainfall in the region is accumulated (Marengo et al., 2015). In hydrological terms, 2265 km$^2$ of drainage area into the system historically generates an annual mean tributary discharge of 38.74 m$^3$/s. Structurally, the system consists of the damming and interconnection of five basins with a useful total storage volume of 988.8 hm$^3$, arranged to transfer water from the Piracicaba River Basin to the Upper Tietê Basin (Fig. 1). As a result, the system had been configured to supply water to about 11 million people in the SPMR before the last acute water crisis in 2013-15 (De Andrade, 2016; Marengo et al., 2015; Nobre et al., 2016; Nobre and Marengo, 2016; PCJ/Comitês, 2016, 2006).

[Figure]

**Figure 1.** System structure composition and catchment areas "Cantareira System": Jaguarí-Jacareí, Cachoeira,
Atibainha and Paiva Castro watersheds. Panel A: Discharge gauge stations; Panel B: rainfall gauge stations; Panel
C: Meteorological gauge stations and Panel D: Centroid of the Eta-INPE grid.

Previously in the SPMR, some severe water shortages were recorded. The first one was during
1953-1954, then from 1962-1963 (Nobre et al., 2016), which apparently motivated the
construction of the Cantareira System and the latest one was from 2000-2001 (Cavalcanti and
Kousky, 2001). Thus, the system, designed to supply the increasing demand for water in the
SPMR, began its partial operation in 1974 and its construction was completed in 1981 with a
30-year permit to transfer up to 35 m$^3$/s according to a periodic technical assessment (Mohor
and Mendiondo, 2017). The Cantareira System is currently administered by SABESP, which
mainly operates the water network in the SPRM, and the Government of Sao Paulo state is its
main shareholder.

However, various studies have identified changes in rainfall trends and temperature extremes, showing an increase in the intensity and frequency of days with heavy rainfall and longer duration of hot dry periods between rainfall events in South America and southeast Brazil (Chou et al., 2014b; Dufek and Ambrizzi, 2008; Haylock et al., 2006; J. A. Marengo et al., 2009; Jose A. Marengo et al., 2009b, 2009a; Nobre et al., 2011; Zuffo, 2015). Although historically, the SPRM study area is not affected by droughts of the same order of Northeast Brazil, the SPRM is progressively becoming vulnerable to water shortages. Therefore, during the recent period of the acute crisis 2013-2015, SABESP undertook reactive measures to control the consumption in the SPMR, such as (Marengo et al., 2015): programmed water cut-offs; bonuses and penalties to reduce and increase consumption, respectively; extraordinary increases of water tariff prices; network pressure reduction; water use from the reservoirs´ dead volume; social awareness campaigns to inform people about shortages; and water distributed by tankers in the most critical areas of the city to provide the Basic Water Requirement (BWR) for human needs. Nevertheless, according to SABESP, there is a slow system recovery, which enables the reestablishment of pre-crisis supply levels (SABESP, 2018a).

**3. Methodology**

The methodology was structured in three modules that are summarized in Figure 2. In the first module, the hydrological simulation was approached by the Water Evaluation and Planning tool (WEAP) (Yates et al., 2005a). The model was calibrated and validated based on the available historical hydrometeorological information (2004-2015) for the study area. Then, from the calibrated hydrological model and the RCM Eta-INPE historical period datasets, the base discharge scenarios were estimated.

In the second module, following the Threshold Level Method (TLM), the "threshold" had to be defined according to stationary and non-stationary assumptions of water demand in the SPMR. Afterwards by analyzing the duration series and extreme deficits through GEV (Generalized Extreme Value) distribution, the Severity-Duration-Frequency curves (SDF) were developed to calculated the intra-annual deficit (J. H. Sung and Chung, 2014). To complete the second module, the average water price and the Cantareira system robustness analysis is defined per each cubic meter of deficit (Mens et al., 2015), as a function of the supply warranty time during the hydrological drought events, to configure the baseline analysis scenarios.

The final module evaluates the Water Utility Company economic profit losses through the baselines scenarios, under the hydrological scenarios developed with the model WEAP, driven by the Eta-INPE RCPs scenarios (2007-2040, 2041-2070, 2071-2099), previously processed by the TLM approach. It should be clarified that, for the analysis under the non-stationary assumption, the growth of water consumption is represented in each projection time step, that is, 2005-2040 corresponds to 31 m$^3$/s, 2041-2070 corresponds to 38 m$^3$/s and 2071-2099 corresponds to 43m$^3$/s.

[Figure]

**Figure 2.** Methodology flowchart and main inputs.

The results, presented as water utility company revenue losses were developed from a set of potential scenarios involving climate uncertainty, human triggering factors and the prediction under extreme theory (Baumgärtner and Strunz, 2014; Wanders and Wada, 2015). The methodology sought to provide a planning water-security support analysis in areas highly dependent on surface water resources.

**3.1. Hydrological projections**

Currently the RCM Eta-INPE (Brazilian National Institute for Space Research) plays an important role in providing information for local impact studies in Brazil and other areas in

South America (Chou et al., 2014b). In order to assess the uncertainties of climate change impacts, the simulation results of the Eta-INPE model were used in this paper. The model is nested within the GCMs MIROC5 and HADGEM2-ES, forced by two greenhouse gas concentration scenarios (RCPs) 8.5 and 4.5 [W/m$^2$] used in AR5 (IPCC 5th Assessment Report); with a horizontal grid size resolution of 20 km x 20 km and up to 38 vertical levels through 30 years of time slices (periods) distributed as follows: 1961-2005 (as the baseline period), 2007-2040, 2041-2070 and 2071-2099 (Chou et al., 2014a, 2014b; Prudhomme et al., 2014). The climate projections of the Eta-INPE model were used to drive the WEAP Rainfall Runoff Model-soil moisture method (World Bank, 2017; Yates et al., 2005a). The WEAP, is an integrated water resource planning tool used to develop and assess scenarios that explore physical changes (natural or anthropogenic) and has been widely used in various basins throughout the world (Angarita et al., 2018; Bhave et al., 2014; Esteve et al., 2015; Foster and Brozovic, 2018; Groves et al., 2008; Howells et al., 2013; Hund et al., 2018; Mousavi and Anzab, 2017; Psomas et al., 2016; Purkey et al., 2008; Vicuña et al., 2011; Vicuna and Dracup, 2007; Yates et al., 2005b). Climate-driven models, such as WEAP provide dynamic tools by incorporating hydroclimatological variables to analyze, in this case, a one-dimensional, quasi physical water balance model, which depicts the semi-distributed hydrologic response through the surface runoff, infiltration, evapotranspiration (Penman-Monteith equation), interflow, percolation and base flow processes (Forni et al., 2016).

The hydrological model comprises 16 sub-basins with a spatial resolution ranging from 67 to 272 km$^2$ (see Figure 1), which defines the natural discharge produced by the Cantareira System. The observed hydrologic data (discharge and rainfall) were taken from HIDROWEB (the National Water Agency database [ANA]), SABESP and the São Paulo state Water and Electricity Department [DAEE]. A network of 52 rain gauge stations and 11 discharge gauge stations were configured, with inputs and outputs by a monthly time-step. On the other hand, the meteorological data from 14 gauging stations (temperature, relative humidity, wind speed and cloudiness fraction) were taken from the National Institute of Meteorology and Center for Weather Forecasting and Climate Research (CPTEC) databases (see Figure 1: panels A, B, C and D). For the basin characterization, we adopted the soil map from (De Oliveira et al., 1999) (1:500,000) and the land use map of 2010 from (Molin et al., 2015) (1:60,000).

The hydrological model was calibrated using a mixed calibration process. A first approximation of the calibration parameters was made by the Model-Independent Parameter Estimation & Uncertainty Analysis software (PEST), automatic calibration tool in WEAP (Doherty and

Skahill, 2006; Seong et al., 2015; Skahill et al., 2009; Stockholm Environment Institute (SEI), 2016), and then the calibration parameters were refined using a manual adjustment technique. In the modeling process, a two-year warm-up period from 2004 to 2005 was established, for the calibration period from January 2006 to December 2010 and from January 2011 to August 2015 as the validation period. Although more extensive periods of calibration and validation are suggested to better represent hydrological dynamics (Gibbs et al., 2018), the absence of observed data restricted the extension of assessment periods. During this process, the following variables were calibrated: Kc (Crop Coefficient), SWC (Soil Water Capacity), DWC (Deep Water Capacity), RZC (Root Zone Conductivity), DC (Deep Conductivity) and PFD (Preferential Flow Direction). The objective functions to measure model performance, widely used in hydrologic applications, were the Volumetric Error Percent Bias (PBIAS), Standard Deviation Ratio (SDR), Nash-Sutcliffe Efficiency (NSE), NSE of the logarithmic of discharges ($NSE_{Log}$) which is more sensitive to low-flows, Coefficient of determination ($R^2$) and Volumetric Efficiency (VE), where the joint maximization of the $NSE_{Log}$ and PBIAS criteria was the objective function to measure model performance (Muleta, 2012).

The WEAP model was calibrated based on eleven discharge gauge stations (see Figure 1, panel A) from the ANA-HIDROWEB dataset (www.ana.gov.br), four of these located at the reservoir entrance of each sub-system (Jaguarí-Jacareí, Cachoeira, Atibainha and Paiva Castro). Cantareira's reservoirs (sub-systems) were set up as a single Equivalent System (ES), where the specific water demands are considered (ANA and DAEE, 2004; PCJ/Comitês, 2006). This ES can be expressed as follows:

$$ES_{Cantareira} = \sum_i^n QN_i - \sum_i^n WD_i \qquad \text{Equation 1.}$$

Where $ES_{Cantareira}$ is the available water for withdrawal from the system, $QN$ is the natural discharge from the reservoir $i$ (sub-system) and $WD$ is the specific water demand in each reservoir (such as the reservoir downstream urban supply).

It is worth noting the sub-basins areas in this case are smaller than each cell of the adopted climate model (400 km²) and although RCMs are an alternative to downscale the coarse resolution GCM, often RCM outputs deviate from the observed climatological data (Kim et al., 2015; Liersch et al., 2016; Smitha et al., 2018a). Therefore, in order to adjust the RCM output dataset, the projections of the Eta-INPE scenarios had to be spatially relocated and bias corrected from observed historical climate conditions (rain and temperature). For this, the "Additive Corrections and Scaling" method was used, a simple approach that assumes the relative mean biases between observed data and model projections (Maraun and Widmann, 2018; Smitha et al., 2018b). The hydrological discharge projections 2007-2099 forced by GCMs and RCPs scenarios can be seen in Appendix B (Fig. B-1).

**3.2. SDF curve development**

Following the flowchart of Figure 2, the Threshold Level Method (TLM) is traditionally used to estimate hydrological drought events from continuous discharge time series (Wanders et al., 2017). TLM was originally called 'Crossing Theory Techniques" and it is also referred to as run-sum analysis (Hisdal et al., 2004; Nordin and Rosbjerg, 1970; Şen, 2015). Usually, different criteria may be used to define the threshold in hydrological drought analysis by the TLM approach (Rivera et al., 2017; Şen, 2015; Tosunoglu and Kisi, 2016). In this study, two monthly desired-yield thresholds were implemented. They were defined from the pre-established water demand in the system (Hisdal et al., 2004; J. H. Sung and Chung, 2014). Initially, a stationary demand (SD) of 31 $m^3$/s was defined as the historical average demand and another non-stationary demand (NSD) of 31 to 42 $m^3$/s over time was defined as a hypothesis representative of the population growth in the SPRM (see Figure 3). These water demand values are consistent with the ANA/DAEE, 2004 study, according to the record and projection scenarios of the population growth of the IBGE[1]. On the other hand, the continuous discharge series were defined from the hydrological modelling result based on Eta-INPE historical dataset (baseline period 1962-2005). From the results of the TLM approach in the Cantareira System, the baseline (historical) scenario, based on Eta-MIROC5 model simulations, showed the greatest hydric deficit under the two water demand scenarios analyzed (SD and NSD), see Figure 3.
* * *
[1] Brazilian Institute of Geography and Statistics: http://www.ibge.gov.br/home/

[Figure]

**Figure 3.** TLM Evaluation from historical discharge WEAP-Eta (base line scenarios), under Stationary (SD) and Non-Stationary Demand (NSD) assumptions as the "threshold level": a. 31 m³/s and Eta-MIROC5. b. 31 m³/s and Eta-HadGEM. c. 31 to 42 m³/s and Eta-MIROC5. d. 31 to 42 m³/s and Eta-HadGEM. Total River basin 2265 km².

Based on the time series of "severity" (or deficit, in m³) and duration (days) in the Cantareira System, obtained from TLM evaluation of the Eta-INPE historical scenarios (see Figure 3), the SDF curves were constructed. To estimate the return periods of drought events of a particular severity and duration, the block maxima GEV frequency analysis distribution was used. In this case, the GEV distribution is useful because it provides an expression that includes all three types of extreme value distributions (Tung et al., 2006).

In various studies addressing SDF curve development, the GEV distribution was consistent with the data sets of extremes, where distributions that use three parameters were required to express the upper tail data (J H Sung and Chung, 2014; Svensson et al., 2016; Todisco et al., 2013; Zaidman et al., 2003). On the other hand, it is suggested that for other durations of drought, other probability distribution functions can be explored (Dalezios et al., 2000; Razmkhah, 2016). However, in this study we took advantage of the versatility of the GEV distribution, considering its flexibility to fit a set of data through the expressions:

$$F(x) = exp\left[-\left\{1 + \xi\left(\frac{x-\mu}{\sigma}\right)\right\}^{1/\xi}\right] \quad \xi \neq 0 \qquad \text{Equation 2.}$$

$$F(x) = exp\left[-exp\left(-\frac{x-\mu}{\alpha}\right)\right] \quad \xi = 0 \qquad \text{Equation 3.}$$

Where the cumulative distribution function *F(x)* depends on μ as a location parameter, α as a scale parameter and ξ as a shape parameter. Therefore, if, μ+α/ ξ ≤ x ≤ ∞ for ξ < 0 is a Type III (Weibull), −∞ ≤ x ≤ ∞ for ξ = 0 is a Type I (Gumbel), and −∞ ≤ x ≤ μ +α/ ξ for ξ > 0 is a Type II (Frechét) distribution (Stedinger et al., 1993).

In order to fill a considerable number of events per interval, droughts were classified into four time intervals from 0 to 31, 0 to 90, 0 to 180 and 0 to 365 days. Thus, considering the adoption of the GEV distribution, the model parameters ξ, α and μ for cumulative durations defined and return periods of 2, 10 and 100 years were estimated using the Method of Maximum Likelihood Estimator (MLE).

**3.3. Water price and Hydrological drought relationship**

According to the flowchart in Figure 2, drought can be addressed as a somewhat unusual economic phenomenon in that it affects both supply (the source) and demand (users), especially in systems dependent on water from a single source (Moncur, 1987). As expected, episodes of water scarcity pose technical, legal, social and economic problems for managers of urban water systems. Traditionally to overcome these episodes, reservoirs play a key role in water supply and demand management, providing security against hydrological extremes (Mehran et al., 2015). However, when the water deficit intensifies, the structural measures are not enough and they must be accompanied by contingency measures, for example, water price regulation instruments, implemented as an incentive for more efficient use (Mechler et al., 2017) .

In Brazil, each state-owned sanitation company has its own water charging policy, where the vast majority use block tariffs as a pricing policy, including SABESP (De Andrade Filho et al., 2015; Mesquita and Ruiz, 2013; Ruijs et al., 2008). In Sao Paulo State, the tariff policy system is regulated by Decree 41.446/96, also for services provided by SABESP. For the water tariff setting, several factors are taken into account, such as service costs, debtors forecast, expenses amortization, environmental and climatic conditions, quantity consumed, sectors and economic condition of the user (SABESP, 1996). These sectors are divided into residential, industrial, commercial or public, and the value that is charged for the service is always progressive. In other words, there is a standard minimum consumption with a fixed value and, based on that, such factors vary the consumption ranges (SABESP, 2018b). From the total water withdrawn from the Cantareira System, urban use is predominant in SPRM, where approximately 49% of the total is for household needs, 31% for industrial needs and 20% for irrigation (Consórcio/PCJ, 2013). In this study, we consider the water-withdrawal for domestic and industrial use in the SPMR, due to the direct dependence of these sectors on the SABESP water supply network, as well as the supply priority that the domestic sector have according to

Brazilian law during drought periods (Lei Nº 9.433 do GOBERNO DO BRASIL, 1997).

Figure 4 shows the TLM analysis with a constant threshold under the same discharge scenario (SABESP 2000-2016), differentiated by the monthly and annual accumulation of the variable.

The monthly step represents the system's natural discharge without regulation ("a" in Figure

5), while the regulated discharge is represented by the annual aggregation of monthly natural discharges ("b" in Figure 5). Assuming this, without the reservoir system ("a" case), with direct water withdrawals (Threshold = 31 m$^3$/s), the average accumulated deficit over these 17 years would be 225% greater than with the reservoir system implemented ("b" case).

The TLM analysis (Figure 5) showed two hydrological drought periods in 2000-2003 and 2010-

2015: one with a lower and another with a higher deficit, respectively. While for the period from 2004 to 2009, a series of smaller droughts in both magnitude and frequency could be overcome by the reservoir system. On the other hand, in 2010-2015, the accumulated deficit, under the regulated scenario, would exceed the useful storage in 70% while for the period 2000-

2003, the accumulated deficit only reached 43% of the system's useful storage capacity.

Therefore, it is clear that over a long period of deficit or strong multi-year droughts, the storage system could be accompanied by other contingency complementary measures.

[Figure]

**Figure 4.** TLM analysis under two time step assumptions during, 2000-2016 period and Threshold equal to 31
m$^3$/s. a) Monthly discharge and b) Annual discharge.

Urban drought management programs incur costs that must be assumed to overcome the water crisis with equity (Molinos-Senante and Donoso, 2016). SABESP in the SPMR, for example, through price-based policies[2] controlled the consumption rates of water users when the hydrological deficit scenarios were presented in the Cantareira System (Iglesias and Blanco, 2008; Millerd, 1984; SABESP, 2018c, 1996). Therefore, during the 2014/2015 drought in SPRM, reactive economic contingencies were implemented, such as increased water tariffs, extra fees and price incentives, which had a detrimental effect on the company's profit margin, which provides the water resource (GESP, 2016; SABESP, 2017a, 2016a).

However, the financial exposure does not always exhibit a strong correlation with weather indices (Zeff and Characklis, 2013). We established a drought revenue loss cost estimation based on the Market Price method (Meyer et al., 2013). To do this, we developed an empirical relationship between the water price (impacts) and drought duration (severity) (Bachmair et al., 2016; Grafton and Ward, 2008; Hou et al., 2018; Mens et al., 2015). Based on the TLM approach, the monthly discharge time series and a constant threshold (31 m3/s) from 2000 to 2018 (Figure 5) was analyzed; aiming to associate the drought characteristics with the adjustment rates of the SABESP database. On one hand, the upper part of Figure 5 shows the drought duration and the annual tariff adjustment with a Pearson correlation coefficient "$r_{xy}$" of 0.481 between them. On the other hand, the lower part represents the deficit volume for each drought duration. In this case, the Pearson correlation coefficient between drought duration and tariff adjustment showed an "$r_{xy}$" value of 0.453.

From the calculated correlation coefficients, a T-student significance test with an alpha of 5% was implemented. Based on the test, it was found that the adjustment rate and the water deficit present a high to medium significance, despite having a lower correlation coefficient. However, in this study the drought duration was assumed as the feature to relate with water price, due to the frequency analysis of the series of annual maxima. Even though the correlation coefficient values showed relatively low values, the use of these drought characteristics may be useful given the lack of information regarding drought and its economic impacts on the study area.
* * *
[2] Database "percentage rate increase" 2001-2018 SABESP:
http://www.sabesp.com.br/CalandraWeb/CalandraRedirect/?temp=4&proj=investidoresnovo&pub=T&db=&doc id=9AA0FF2088FBF0A8832570DF006DE413&docidPai=AB82F8DBCD12AE488325768C0052105E&pai=fil ho10

[Figure]

**Figure 5.** Empirical relationship between Cantareira System drought duration "blue-bar in days" [derived from
monthly average discharge analysis], Cantareira System drought deficit "red-bar in $10^6$-m$^3$" and annual price
adjustment rates under variate hydrological conditions in percentage.

From the recent drought events in SPMR, which significantly affected the water supply i.e in
2000/2001 (Cavalcanti and Kousky, 2001) - 2013/2015 (Nobre et al., 2016) and the TLM
analysis that showed some degree of correlation between annual events and priced adjustment
rates, an empirical system analysis robustness against the drought duration was proposed (Mens
et al., 2015). Our robustness analysis is based on the assumption of the main impacts derived
from water supply problems in the SPRM, which appear to be related with medium to prolonged
duration events and medium to high severity (up to 365 days or two consecutive annual cycles).
Therefore, three priced-adjustment vs. drought duration scenarios were established (see Figure
E-2 in the supplementary material). First, 100% water availability. In this scenario, the
reservoir network is not essential to ensure water supply (drought duration between 0 and 90
days). Second, the water availability with supply warranty and dependence on the storage
system. In this scenario, the reservoir network provides resilience during droughts of smaller
magnitudes and duration (drought duration between 90 and 180 days). Third, stored water
shortage and forced interruption of supply. In this scenario, the water deficit prevails with extra
fees and other savings measures (drought duration between 90 and 365 days).

Since the water price formation study is not part of this work as it entails a complex
microeconomic analysis (Garrido, 2005), we adopted the average prices of water (Bulk Water
Tariff, 2016) in the SPMR, for the Domestic and Industrial sectors (SABESP, 2016b).
Therefore, based on the previous analysis (Figure 6), the following was adopted: First, during
the most severe droughts, an increase in the water tariff for the following period is expected.
Second, on the contrary, when the smaller deficits are overcome with the water stored in the
system, the increase in tariffs is a consequence of the annual Consumer Price Index (CPI) and other tariff updates according to the law (SABESP, 2016b). Thus, the approach requires some additional assumptions explained as follows:

i.   Based on the current average prices for the domestic and industrial sectors, a base-water-price was established to analyze US$ 3.38 per $m^3$, assuming that this value is given considering normal supply conditions or 100% water availability, ii.  From the SDF curve construction intervals (cumulative drought duration) and three class intervals of the annual tariff adjustment (min. 6% to max. 17%, see Figure E-1 in appendix E), the water prices were established (see Table 1).

**Table 1.** Main assumptions for establishing the tariff water price according to the drought duration.

| Drought Duration Interval (days*) | Water Tariff Adjustment adopted (%) | Average price (US$/ $m^3$) | Cantareira System robustness characteristics scenarios |
|---|---|---|---|
| From 0 to 31 | 0 | 3.38 | 100% water availability base scenario |
| From 0 to 90 | 6 | 3.58 | 100% water availability |
| From 0 to 180 | 10 | 3.71 | Water availability with storage dependency |
| From 0 to 365 | 17 | 3.95 | Water deficit (multi-year droughts) |

**\* Cumulative drought duration**

Table 1 represents the inelastic behavior of the Price Elasticity of Demand (PED) showing closer intervals as water supplies are reduced due to drought and higher prices imposed to try to reduce demands. Hence a successful price-based rationing policy requires a progressive increase if the demand becomes predominantly inelastic (Mays and Tung, 2002), as the proposed hypothesis establishes in this case. More studies of price elasticity and water scarcity can be found in (Freire-González et al., 2017; Mansur and Olmstead, 2012; Ruijs et al., 2008).

The final step of the methodology (see Figure 2) defines the calculation of the drought impacts through the management horizons (2007-2040, 2041-2070 and 2071-2099). This calculation was carried out for the cumulative drought periods greater than 180 days, considering that from this duration, the supply begins to show an important dependence of the Cantareira reservoir System.

**4.  Results and discussions**

**4.1. Hydrological modeling**

The hydrological model structure performed in monthly time steps and was calibrated-validated following the described procedure in Section 3.1. To improve the calibration procedure, multiple statistical evaluation criteria were used (Gibbs et al., 2018; Kumarasamy and Belmont, 2017). This is important because analyzing multiple statistics can provide an overall view of the model based on a comprehensive set of indexes on the parameters representing the statistics of the mean and extreme values of the hydrograph (Moriasi et al., 2007).

The equivalent system hydrograph for calibration and validation periods are shown in Figure 6. The colors in Figure 6 represent the classifications suggested by (Moriasi et al., 2007) and are as follows: green for "very good" (NSE > 0.75; PBIAS < ±10%; RSR < 0.50), yellow for "good or satisfactory" (0.75 > NSE > 0.5; ±10% < PBIAS < ±25%; 0.50 < RSR < 0.60), red for "unsatisfactory" (NSE < 0.5; PBIAS > ±25%; RSR > 0.70). Moreover, the correlation coefficient ($R^2$) and the VE criterion values close to 1.0 mean that the prediction dispersion is equal to that of the observation (Krause and Boyle, 2005; Muleta, 2012). It is important to note that in the validation period (2011-2015), most of the recent drought event were simulated with an acceptable performance, although there is a tendency to overestimate periods of low flow.

[Figure]

**Figure 6.** WEAP Hydrographs Cantareira Equivalent System (ES) performance criteria for Calibration (2006 - 2010) - Validation (2011 – 2015) periods. The calibration and validation performance criteria for each sub-basin in the system can be found in the "Complementary Material" - Appendix A. – Table A-1.

Individual watershed hydrological modelling performance ratings are presented in Appendix-A, Table A-1. Moreover, several statistical criteria were considered in the evaluation of the calibration process, where each criterion covers a different aspect of the resulting hydrograph. Five sub-basins were modeled within the Jaguarí-Jacareí sub-system (Sub B-F28, B-F23, B-F25, Jaguarí and Jacareí). This five sub-system represents approximately 46% of the total available water and showed the best modelling performance statistics, compared to the other subsystems.

**4.2. SDF curves**

Using the traditional frequency analysis, the severity-duration-frequency curves for two threshold levels and two discharges (from Historical_RCMs WEAP outputs) were developed as shown in Fig. 7. For the SDF curves configuration, the Generalized Extreme Values (GEV) function was used. It can, therefore be observed from the SDF results that according to the fit data set (Appendix C), the shape parameter ($\xi$) varies with the drought duration, therefore for a drought interval of more than 180 days, the Probability Distribution Function (PDF) Type I presents a better fit, even for the two proposed demand scenarios. On the other hand, droughts with duration intervals of less than 90 days, under stationary and non-stationary demand scenarios, had a better fit to FDP Type III (see Tables D-1 to D-4 in Appendix D). Moreover, the fit diagnostic plots "Empirical quantile vs Model quantile" (QQ-plot) and "Return level vs Return period" (RR-plot) show the relationship between the model, the data fit and prediction capacity (Appendix C). Therefore, in terms of the quantiles, the QQ-plot shows the data trend to follow the model line in most cases. While the predictive capacity of the model, represented by the RR-plot, shows a decrease as the return period increases.

[Figure]

**Figure 7.** SDF curves under stationary and non-stationary demand assumptions and historical discharge WEAP-Eta scenarios: a. (SD) 31 m³/s and Eta-MIROC5. b. (SD) 31 m³/s and Eta-HadGEM. c. (NSD) 31 to 42 m³/s and Eta-MIROC5. d. (NSD) 31 to 42 m³/s and Eta-HadGEM.

Based on the relationship between the Cantareira System Drought-Cost-Robustness curve (see details in Figure E-2 supplementary material) and the SDF curves (see Figure 7), the base functions of Severity-Duration-Impact of drought were built to estimate the base-line scenarios of damage cost in the water utility company. These scenarios are shown in Figure 8, under different recurrence events (Rp scenarios), climate projections (RCPs – RCMs) and demand variability scenarios (SD – NSD). Each pair of lines in Figure 8 (continuous and dashed), show a range of uncertainty associated with the considered change drivers.

[Figure]

**Figure 8.** Severity-Duration-Impact curves. Sector **a.** Severity-Duration-Frequency-Profit Loss under the historical *Eta-MIROC5* scenario. Sector **b.** Severity-Duration-Frequency-Profit Loss under the historical *Eta-HadGEM* scenario. Note: *SD* and *NSD* are the stationary or non-stationary demands, respectively; "*VD*" is the volume deficit, under return period of 2, 10 and 100 years; *% of year* is the drought event duration in relation to one year.

**4.3. Economic impacts under climate change**

The results describe the net present value (NPV) of the potential economic impacts produced by hydrological drought durations greater than 180 days. These impacts are presented considering the climate, demand, severity and recurrence scenarios during the analysis periods:

2007-2040, 2041-2070 and 2071-2099. The evaluation of the drought's economic impact in the water company showed in general, revenue losses per analysis period between 0.003% and 0.021% related to the GDP in the SPMR in 2017 (SEADE, 2018). This relatively low range of percentage revenue losses is, in fact, significant for the regional economy since SPMR accounts for approximately 18% of the Brazilian GPD.

Figure 9 shows the economic impacts on the water utility company under an analysis that independently discriminates radiation scenarios (RCP), GCM and water demand. Figure 9 also compares the relative difference between scenarios and time periods, using the median statistic and standard deviation, the latter as a measure of dispersion. In general, the results in Figure 9 reveal that under the driver water demand, the most propitious scenarios are configured for the generation of greater economic impacts (on average), followed by radiation and GCM drivers, respectively. Likewise, in Sector "a", the impacts analyzed under RCP scenarios 4.5 and 8.5 showed a low difference percentage in variability and median. This can be explained from the study by Chou et al. 2014, where the Eta-INPE results establishes that, in the future, there is no clear trend in the average precipitation and during the summer, the time series show a trend for a reduction in precipitation in both emission scenarios, RCP 8.5 and 4.5. While for sector "b" (RCM), the outputs nested in Eta-MIROC5 presented higher revenue losses in the company than those based on HadGEM-ES. This difference can be attributed to the annual cycle of precipitation, which shows that the ETA-INPE simulations driven by MIROC5 produces generally less precipitations during the dry season, therefore the water deficit during this period will be more critical (Chou et al 2014a). Finally, sector "c", where the Non-stationary demand (NSD) trend imposed the larger differences in the magnitude and variability percentage impacts (human influences), suggesting that the demand-related (population growth) uncertainty would be far greater than that associated with climate sensitivity.

[Figure]

**Figure 9.** Impacts and relative differences between scenarios in median 50th percentile (Med.) and standard deviation (σ). Sector "a": Impacts based on RCP scenarios. Sector "b": Impacts based on RCM scenarios. Sector "c": Impacts based on demand scenarios. Through analysis periods, Orange (2007-2040), Green (2041-2070) and Yellow (2071-2099).

Under a different grouping configuration for the analysis of the results (see Figure 10), the impacts assessment was conditioned by the scenarios joint study of climate forcing (Eta-GCM) and radiation (RCP). Based on this scheme, it was found that the largest economic impact was represented by the Eta-MIROC5_4.5 climate-forcing scenario, while smaller impacts (on average) were observed in the Eta-HadGEM-ES_4.5 scenario. In addition, the Eta-MIROC5 scenario showed the maximum values of the median 50th percentile (Max.Med.) and standard deviation (Max.SD) between the set of time period panels, which concludes that the climate forcing based on the MIROC5 model is the main driver of the impacts and variability between
analyzed climate drivers (GCM).

[Figure]

**Figure 10.** Economic impacts comparison between Eta-INPE_RCP_GCM based scenarios throughout the projection time period: first panel 2007-2040, second panel 2041-2070 and third panel 2071-2099.

Figure 11 describes a third results analysis scheme. In this case, the impacts are evaluated based
on the return periods. In the box plot of Figure 13, an increasing tendency of the dispersion is
observed in the measure in which the projected time horizon is more distant, probably due to
the greater uncertainty in future climate projections (Cubasch et al., 2001). On the other hand,
the higher periods of return reflect impacts of greater magnitude, as expected.

In all cases, the average economic impacts projected for the period 2040-2071 presented lower
values compared with the other two periods analyzed. According to a study by Lyra et al. (2017)
in which the most recent Eta-INPE model simulations were performed at more detailed scales,
the annual total precipitation (PRCPTOT) and maximum number of consecutive days with
precipitation (CDD - CWD) indexes for the Sao Paulo region showed better results in terms of
favoring water availability during this period. On the contrary, the period 2007-2040 presented
the greatest economic impacts (evidence of the recent water crisis) with the lowest dispersion
(less uncertainty) in relation to the other projected time periods. While the projection of the
2071-2099 period showed an impact magnitude close to the 2007-2040 period, given that both
Eta-INPE simulations intensify the reduction of precipitation toward the end of the century in
Southeast Brazil, with an annual rainfall reduction above 40% and a reduction of precipitation
extremes (Chou et al., 2014a; Lyra et al., 2017).

[Figure]

**Figure 11.** Drought impact variability between return period scenarios during the projection periods: first panel 2007-2040, second panel 2041-2070 and third panel 2071-2099.

**4.4. Considerations on Uncertainties**

The methodology adopted here includes a model chain, typical in exercises of hydrological regime projection through hydrologic modelling under climate change projections (Fowler et al., 2007; Jones, 2000; Wilby and Harris, 2006). This model chain incorporates several sources of uncertainty such as those listed by Honti et al. (2014) and Jobst et al., (2018): 1) the climate model; 2) the downscaling method or an RCM application, the latter as in our work; 3) the hydrological model, and (4) the inherent modeling uncertainty of coupling different climate-hydrology spatiotemporal scales.

In this case, the systematic analysis of change drivers (uncertainty sources) offers a set of results around potential scenarios to frame uncertainty (Refsgaard et al., 2007; Rodrigues et al., 2015) while the drivers sensitivity analysis is proposed as a part of the results in this study. Montanari (2007), however, advocates that some methods commonly used for uncertainty assessment do not address uncertainty, but only model sensitivity. Moreover, although some studies indicate that the climate projections surpass the hydrological uncertainties (Bates et al., 2008; Nóbrega et al., 2011), Honti et al. (2014) reinforces that different methods of uncertainty assessment may lead to different conclusions.

Our methodology also included a drought indicator development, through the TLM approach, demand scenarios and a drought cost estimation based on the Market Price method. (Hou et al., 2018; Mens et al., 2015). Results showed that drought deficits are influenced not only by the modeled inflows at a lumped scale, throughout the period of 2007-2099, but also in our case study by the reservoir operation. In fact, the spatially-combined operation of existing reservoirs may be different from our considerations, adopting an "equivalent system" (ES) without a future layout change. On the one hand, the system demand scenarios are based on current (historical period 2004-2016, (SABESP, 2017b)) best knowledge information, and the adoption of two scenarios aimed at giving a broader, realistic view of the different possible outcomes due to expected population growth (ANA and DAEE, 2004; IBGE, 2018). On the other hand,
the economic loss estimation, based on the aforementioned drought event measures, does not
incorporate eventual market changes, currency changes or even subsidies. Conversely, our loss
estimation assumes that those economic measures, i.e. water tariff adjustments were and would
continue being adopted by the water utility, as a trigger determinant once the drought hazard
happened. Because this triggering factor would temporarily occur either promptly or slowly,
when structural measures were not enough to secure water supply under eventual hydro-
meteorological conditions and water demand, uncertainties in cost analysis could increase.

**5. Conclusions and recommendations**

This paper developed a methodology with application to assess economic impacts of drought
risks for water utilities through an assessment under climate and water demand scenarios. In
this example, the SDF framework has linked climate, hydrology and economy factors, using
Sao Paulo Metropolitan Region dependence on the Cantareira Water Supply System, Brazil. In
this paper, we consider these results preliminary, but with valuable information for a water
utility interested in the drought risk losses.

Methodologically, first we characterized the hydrological droughts through the SDF curves,
from the hydrological modeling by the baseline period of the RCM. Second, the SDF was
coupled with a local water demand development based on the supply warranty time percentage
during the drought events. Under these assumptions, an empirical drought economic impact
curve was setup, representing the Water Utility Company profit losses due to the impossibility
of supplying demand during hydrological drought periods. Additionally, our results could elicit
further implications for drought risk reduction and management.

The main results of the methodology implemented were: the great financial vulnerability of the
water utility company of the SPMR against the hydrological drought. Possibly the maximum
supply capacity of the system is reaching its limit due to the growing demand and the new
challenges represented by climate change. The main driver of economic impacts under the
analysis scheme turned out to be the water demand dynamics. The Eta-MIROC5 scenario
proved to be primarily responsible for the economic impacts compared to other climate drivers.
Comparatively, the RCP 4.5 and 8.5 radiation scenarios showed no major difference between
them. The scenario of projected impacts for the period 2071-2040 showed the greatest
dispersion among time scenarios, while the closest scenario 2007-2041 showed less dispersion
and greater impacts on average. The WEAP model proved to be a versatile tool for the construction-calibration-validation process of the model, when it is implemented in climate impact studies. The approach for the characterization of the drought "TLM" showed to be a tool easily applicable to describe quantitative changes in hydrological drought during long periods with change in water demands (Thresholds).

On one hand, this SDF framework could help analyze the impacts from key drivers, such as climate, land use and water withdrawal rates in complex or recurrent drought patterns. In addition, this SDF framework could link interdisciplinary studies, with broader relationships in relation to water security, energy security and food security. Thus, we recommend future research of the SDF framework linked to: Palmer's drought indices (Rossato et al., 2017); a model-based framework to disaster management (Horita et al., 2017); an ecosystem-based assessment for Eco-hydrological modelling (Taffarello et al., 2018); effectiveness of drought securitization under climate change scenarios (Mohor and Mendiondo, 2017). Moreover, the SDF framework is capable of integrating actions towards: dynamic price incentive programs related to wise human-water co-evolution patterns; water-sensitive programs under deep cultural features; socio-hydrological observatories for water security; feasibility analysis of the economic impacts of implementing new technologies for water economy and flow measurement; leakage control; detecting and legalizing illegal connections and water reuse, among others. Furthermore, dissimilarities pointed out from climate scenarios (see Figure 11) would suggest a set of possibilities to face the uncertainty. For instance, the SDF framework would guide the decision-making of water utility profits to cope with economic impacts of drought risks in the long and medium term. In addition, the expected profit loss over the long-term would serve as the initial estimate for financial contingency arrangements as insurance schemes or community contingency funds. In general, the SDF framework developed here can be proposed as a planning tool to mitigating drought-related revenue losses, as well as being useful for the development of water resource securitization strategy in sectors that depend on water to sustain their economies.

The following should be considered for further studies to strengthen decision-making based on results of the tool: despite having achieved an acceptable performance, the inclusion of more gauge stations could not only improve calibration performance but also cover a larger sample space of events, increasing the confidence of projections. Introduce a direct measure of the economic impacts resulting from multi-year deficits of annual duration not entire, although, the methodology can assimilate multiple consecutive years and entire deficits, the cumulative impacts would be underestimated. On the other hand, in order to have a methodological comparative standard, more regional studies of SDF curves need to be implemented, considering the spatialized analysis and broader statistics methods. Finally, it is a fact that the reliability of SDF curve estimates depend on the quality and extent of the records used, or in this case, the capacity of regional climate models to reproduce the observed distribution of extreme events.

**Acknowledgments**

The authors would like to thank the support from several agencies in Brazil and Colombia: the Administrative Department of Science, Technology and Innovation (COLCIENCIAS) Doctoral Program Abroad No 728-2015, CAPES-PROEX-1650/2017/23038.013525/2017-30, CAPES Pró-Alertas #88887.091743/2014-01, CNPq #307637/2012-3, CNPq #312056/2016-8 PQ and CNPq #465501/2014-1 and FAPESP 2014/50848-9 Water Security of the INCT-Climate Change II. The Sao Paulo State Water Utility Company, SABESP, kindly provided relevant information for this study. All co-authors declare no conflict of interest. The third author thank to Coordination of Superior Level Staff Improvement (CAPES) and to the Programme of Postgraduate in Hydraulics and Sanitation (PPG-SHS) for the postdoctoral fellowship.

Detailed information is available in the Supplementary Material.

[revised manuscript text omitted]

**Figure E-2.** Cantareira System Drought-Cost-Robustness curve, based on the water price and drought duration. The supply warranty time is a defined index for the construction of the drought impact curve. In this case, the draught impact curve describe the relationship between the duration of the drought (supply guarantee time), the water price adjustment rate and the system robustness. Supply warranty time is the ration between 100% Supply warranty time during 31 days and the Analysis Scenario of Supply warranty time (days). For example, 31 days/31days=1; 31days/90days=0.34; and 31days/180days=0.17 and 31 days/365 days=0.084.